# TS-RAG: Retrieval-Augmented Generation based Time Series Foundation Models are Stronger Zero-Shot Forecaster

**Kanghui Ning[1]    Zijie Pan[1]    Yu Liu[3]    Yushan Jiang[1]**
**James Yiming Zhang[4]    Kashif Rasul[2]    Anderson Schneider[2]**
**Lintao Ma[3†]    Yuriy Nevmyvaka[2†]    Dongjin Song[1†]**

[1]School of Computing, University of Connecticut, Storrs, USA.
[2]Department of Machine Learning Research, Morgan Stanley, New York, USA.
[3]Ant Group, Hangzhou, China.
[4]TWG Global, New York, USA.

## Abstract

Large Language Models (LLMs) and Foundation Models (FMs) have recently become prevalent for time series forecasting tasks. While fine-tuning LLMs enables domain adaptation, they often struggle to generalize across diverse and unseen datasets. Moreover, existing Time Series Foundation Models (TSFMs) still face challenges in handling non-stationary dynamics and distribution shifts, largely due to the lack of effective mechanisms for adaptation. To this end, we present TS-RAG, a retrieval-augmented generation framework for time series forecasting that enhances the generalization and interpretability of TSFMs. Specifically, TS-RAG leverages pre-trained time series encoders to retrieve semantically relevant segments from a dedicated knowledge base, enriching the contextual representation of the input query. Furthermore, we propose an Adaptive Retrieval Mixer (ARM) module that dynamically fuses the retrieved patterns with the TSFM's internal representation, improving forecasting accuracy without requiring task-specific fine-tuning. Thorough empirical studies on seven public benchmark datasets demonstrate that TS-RAG achieves state-of-the-art zero-shot forecasting performance, outperforming the existing TSFMs by up to 6.84% across diverse domains while also providing desirable interpretability. Our code and data are available at:
`https://github.com/UConn-DSIS/TS-RAG`.

## 1   Introduction

Time series forecasting, which aims to predict future values of a sequence based on its past observations, plays a critical role in various real-world applications, *e.g.*, finance [1], healthcare [2], energy management [3], and climate science [4]. Modeling time series data essentially captures the temporal dependency patterns in the form of trend, seasonality, autocorrelation, *etc*, to make accurate predictions and generalize across different datasets. In the past, a substantial amount of effort has been made to tackle this problem. Traditional statistical methods such as AutoRegressive Integrated Moving Average (ARIMA) [5] work well for stationary time series but struggle with complex dependencies and non-linear patterns. Machine learning approaches, such as Random Forest [6] and XGBoost [7], can handle external covariates of features but fail to capture long-range dependencies. Deep learning techniques, including Long Short-Term Memory (LSTM) [8], Gated Recurrent Units (GRUs) [9], Temporal Convolutional Networks (TCNs) [10], Graph Neural Networks (GNN) based models [11, 12], and transformer based models [13, 14] are typically trained within specific domains

---

†Correspondence to: Lintao Ma <lintao.mlt@antgroup.com>, Yuriy Nevmyvaka <yuriy.nevmyvaka@-morganstanley.com>, Dongjin Song <dongjin.song@uconn.edu>.

39th Conference on Neural Information Processing Systems (NeurIPS 2025).

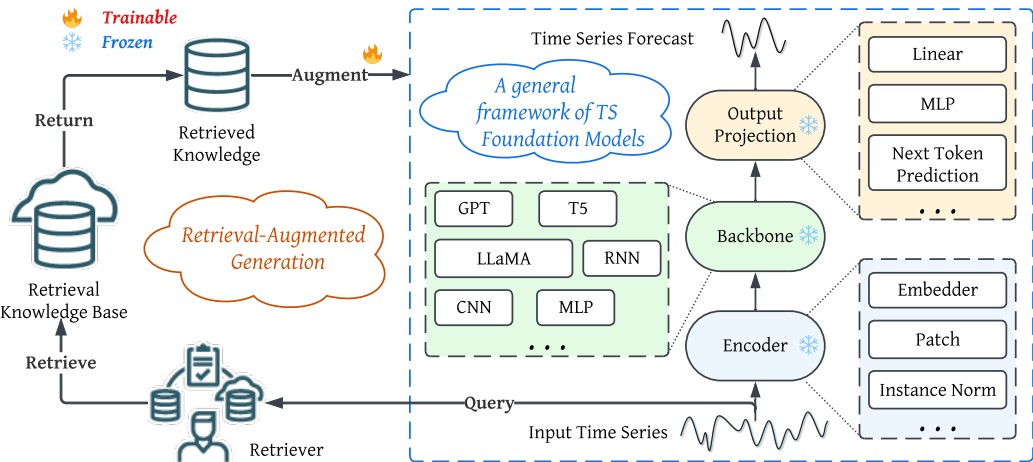

Figure 1: Overview of the proposed TS-RAG framework. Given an input time series as the query, the retriever accesses a knowledge base to obtain semantically related information. The retrieved knowledge is subsequently integrated into a frozen time series foundation model (which can adopt various architectures and design choices) to enhance forecasting performance.

and often struggle to generalize effectively to diverse, unseen datasets. More recently, there is a prevalent interest in adapting Large Language Models (LLMs) [15, 16, 17, 18] for time series tasks [19, 20] and developing Foundation Models (FM) tailored for time series data [21, 22, 23, 24]. Although both LLMs and FM have shown great promise in improving forecasting accuracy and handling complex temporal dynamics, they still face immense barriers when applied to zero-shot forecasting tasks, limiting their real-world applicability.

Specifically, recent work [25] has explored the potential usage of LLMs as zero-shot forecasters for time series tasks. While fine-tuning LLMs can facilitate adaptation and yield better performance over specific datasets, such methods often struggle to generalize across diverse, unseen domains [26, 27, 28], and typically involve substantial computational overhead, even when applied to limited data. Time Series Foundation Models (TSFMs) [29, 30, 31, 32] have emerged as a promising alternative by learning time series representations across a wide range of time series data. However, they lack inherent mechanisms for domain adaptation, as they cannot incorporate external contextual knowledge dynamically. This shortcoming limits their capability in handling complex, non-stationary, and evolving time series patterns. Furthermore, most TSFMs offer limited interpretability, which poses challenges for their deployment in high-stakes or decision-critical applications.

Recently, Retrieval-Augmented Generation (RAG) [33] has demonstrated significant success across various Natural Language Processing (NLP) tasks [34, 35, 36, 37]. By retrieving relevant document segments and incorporating them into prompts, RAG refines the existing prompts and enables LLMs to generate more informed, context-aware outputs, thereby improving both accuracy and adaptability in diverse applications. Motivated by this paradigm, we propose TS-RAG, a retrieval-augmented generation based time series forecasting framework. TS-RAG dynamically retrieves semantically relevant time series patterns and integrates them into the forecasting pipeline, enabling strong zero-shot performance without the need for fine-tuning. In addition, it significantly enhances the interpretability of TSFMs by providing contextual evidence for their predictions. An overview of the proposed framework is illustrated in Figure 1, where a retriever queries a knowledge base for semantically related time-series information, and the retrieved knowledge is used to augment a frozen TSFM that can adopt diverse architectural designs.

Instead of simply relying on the input time series query, TS-RAG first adopts pre-trained time series encoders to retrieve relevant time series segments from a dedicated knowledge database, providing valuable contextual knowledge for forecasting. Next, to effectively integrate retrieved time series knowledge, TS-RAG leverages a learnable Adaptive Retrieval Mixer (ARM) augmentation module, which can dynamically fuse retrieved patterns with the input time series query, ensuring that the model benefits from both existing knowledge and the current query. With retrieval-augmented generation, TS-RAG not only can circumvent the need for fine-tuning on specific datasets but also can

utilize retrieved segments to provide explicit rationales to enhance the interpretability of the model's predictions. Finally, thorough empirical studies on seven public benchmark datasets demonstrate that TS-RAG achieves state-of-the-art zero-shot forecasting performance, outperforming existing TSFMs by up to 6.84% across diverse domains while exhibiting desirable interpretability, highlighting its potential as a robust and generalizable forecasting framework.

## 2 Related Work

**Time Series Foundation Models** Existing LLM-based time series forecasters [26, 38, 39, 27, 40, 28, 41] have demonstrated remarkable achievements in in-domain time series analysis. However, the domain adaptation challenges and significant computational costs associated with LLM-based models have motivated the emergence of time series foundation models as a more efficient and scalable alternative. Inspired by recent advancements in Natural Language Processing (NLP) and Vision Transformers [42], Time Series Foundation Models have rapidly developed and drawn significant attention. These models have demonstrated strong generalization capabilities across diverse datasets, leading to substantial progress in time series forecasting. Lag-Llama [21] and TimeGPT-1 [22] are pioneering forecasting foundation models, pre-trained on extensive time series datasets spanning multiple domains. Lag-Llama utilizes lagged time series features and the LLaMA architecture [43], while TimeGPT-1 adopts an encoder-decoder transformer structure to handle forecasting tasks effectively. Following the paradigm of patching tokenization and optimization, models such as TimesFM [29], MOMENT [44], Timer [45, 46], and Sundial [47] first patch and embed continuous time series values, and subsequently model the output distributions and perform point forecasting. To further improve efficiency, Tiny Time Mixers (TTMs)[30] train a compact foundation model, while TimeMOE[48] adopts a sparse mixture-of-experts architecture that activates only a subset of expert networks during inference, thereby maintaining strong performance with reduced computational cost. However, deterministic predictions usually cannot satisfy the requirement of decision-making. To address this limitation, Moirai [32] trains a probabilistic model that captures a mixture of distributions. Building on language modeling techniques, Chronos [31] discretizes time series through scaling and quantization, and models the resulting categorical distributions using cross-entropy loss. To enhance inference efficiency and forecasting accuracy, Chronos-Bolt replaces discrete tokenization with a patch-based input strategy and utilizes decoder representations to produce quantile forecasts over multiple future steps, achieving improved performance over its predecessor. These methods, however, lack inherent mechanisms to incorporate external contextual knowledge dynamically to facilitate zero-shot learning and suffer from limited interpretability.

**Retrieval-Augmented for Time Series Forecasting** Although both LLMs and TSFMs have achieved strong performance in time series forecasting, they can still struggle in scenarios involving non-stationarity or distribution shifts. To address these challenges, Retrieval-Augmented Generation (RAG) techniques [33] offer a promising approach to incorporate external knowledge and enhance generalization and robustness. Recent works have explored the integration of retrieval mechanisms into time series forecasting. Among them, several approaches rely on fine-tuning to adapt the model to downstream tasks, including ReTime [49], RATD [50], TimeRAG [51], and RAFT [52]. ReTime [49] proposes relational retrieval and content synthesis for retrieval-based time series forecasting. RATD [50] utilizes retrieved historical time series to guide the denoising process of diffusion models, enhancing forecasting performance. RAFT [52] integrates retrieval with a multi-resolution forecasting framework. These methods typically construct the retrieval database from the training set of the target task and require fine-tuning for model adaptation. Similarly, TimeRAG [51] incorporates retrieved sequences into the forecasting process by using a frozen LLM backbone, and introduces a trainable reprogramming layer to align time series and text modalities.

RAF [53] is a pioneering work that introduces a retrieval-augmented framework for **zero-shot** time series forecasting. It utilizes Chronos as the backbone model and constructs the augmented input by directly concatenating the processed retrieved context with the original time series input. However, this approach may face scalability and efficiency challenges when applied to large-scale retrieval databases. Moreover, RAG techniques for time series forecasting remain underexplored, particularly in terms of knowledge base construction, augmentation strategies, and their integration with TSFMs to facilitate zero-shot forecasting. To bridge these gaps, we propose TS-RAG, a retrieval-augmented framework specifically designed to enhance **zero-shot forecasting capabilities of TSFMs**.

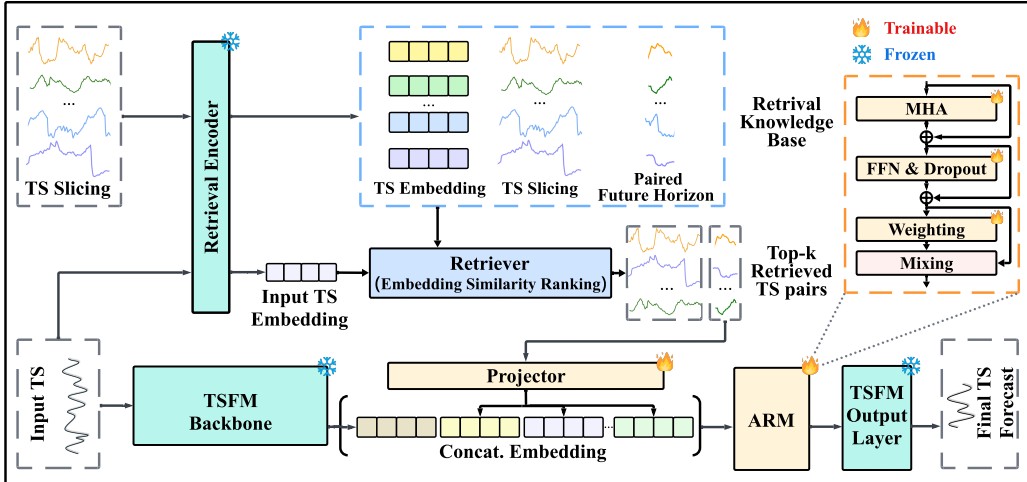

Figure 2: The TS-RAG model architecture processes an input time series by retrieving the top-$k$ semantically similar time series segments and their corresponding future horizons from a knowledge base (via Retriever), based on embedding similarity. These retrieved segments are then integrated with the input series embedding using the proposed Adaptive Retrieval Mixer (ARM) augmentation module, enabling the model to generate the final forecast with enriched contextual information.

## 3 TS-RAG for Zero-Shot Time Series Forecasting

**Overview**: The proposed TS-RAG consists of three key components, *i.e.*, a time series foundation model, a retriever, and a learnable Adaptive Retrieval Mixer (ARM) augmentation module, as shown in Figure 2. Given an input time series, a pretrained retriever encoder first generates the corresponding embedding. This embedding is then compared with time series context embeddings previously stored in the retrieval knowledge base to retrieve the top-$k$ similar time series pairs. Each retrieved pair includes a historical context and its corresponding forecasting horizon, and the forecasting horizon is utilized for augmentation to refine the zero-shot time series forecasting.

The retrieved future horizons of top-$k$ similar time series pairs are first transformed into embeddings and then fed into the ARM augmentation module along with the input time series embedding generated by the TSFM backbone. The ARM augmentation module adaptively assigns importance scores to these embeddings, dynamically integrating them into a unified representation. This final representation is then passed through the output projection layer of the TSFM to produce the enhanced time series forecast. Below, we provide details for the construction of the retrieval knowledge base (Section 3.1), the TS-RAG framework architecture (Section 3.2), and the pretraining strategy used to enable zero-shot inference (Section 3.3).

### 3.1 Construction of the Retrieval Knowledge Base

TSFMs are typically pretrained on a multi-domain time series dataset to enhance their generalization capability. For TS-RAG, we adopt a similar strategy, *i.e.*, construct a multi-domain dataset and specifically focus on learning the ARM augmentation module. We leverage the pretraining dataset of Chronos [31], which utilizes TSMixup to randomly combine time series data points from various domains. This approach enhances data diversity by blending different patterns, thereby improving the model's ability to generalize. Given the fact that the trainable parameters of TS-RAG are significantly less than those in the TSFM backbone and the retrieval encoder (more discussions are available in Appendix B.4), we uniformly sample a subset from the Chronos pretraining dataset to serve as the pretraining dataset for TS-RAG. Based on this subset, we can further draw a subset to construct the retrieval knowledge base for TS-RAG, which will be used in the inference (forecasting) stage.

The time series data stored in the knowledge base is processed into standard pairs, each consisting of a context window and its corresponding forecasting horizon. Formally, this can be expressed as: $\{(\mathbf{x}_i, \mathbf{y}_i) \mid i = 1, 2, \ldots, n\}$ where $\mathbf{x}_i \in \mathbb{R}^T$ is the context window of the $i$-th time series with length

$T$, $\mathbf{y}_i \in \mathbb{R}^L$ is the future horizon of length $L$ associated with $\mathbf{x}_i$, and $n$ is the total number of pairs in the knowledge base. Based on the time series context windows $\{\mathbf{x}_i | i = 1, 2, \ldots n\}$ stored in the retrieval knowledge base, we employ a pretrained retrieval encoder to generate their embeddings $\{\mathbf{e}_i | i = 1, 2, \ldots n\}$ where $\mathbf{e}_i \in \mathbb{R}^d$. These embeddings are then stored along with the corresponding time series data in the knowledge base. As a result, the structure of the retrieval knowledge base can be formally described as the following set of triplets:

$$\mathcal{D} = \{(\mathbf{x}_i, \mathbf{e}_i, \mathbf{y}_i), | i = 1, 2, \ldots, n\} \tag{1}$$

where $\mathcal{D}$ represents the retrieval knowledge base, and $\mathbf{e}_i$ denotes the embedding of the input sequence $\mathbf{x}_i$, computed by a pretrained retrieval encoder, thereby facilitating an efficient training process.

## 3.2 Architecture of TS-RAG Framework

By leveraging relevant time series contexts retrieved from an external knowledge database, TS-RAG can enrich the input query time series with additional contextual information, thereby improving both model generalization ability and forecasting accuracy.

Within TS-RAG, a TSFM has three key components [54]: an encoding layer, which may include normalization and embedding layers to preprocess and transform the input time series; a backbone, typically implemented as a transformer based model (*i.e.* GPT [55], T5 [56], Llama [16] *etc.*) to extract temporal representations; and a projection layer, often implemented as a multi-layer perceptron (MLP), which maps the temporal representations from the backbone to the final prediction values.

TS-RAG introduces two additional components: a **Retriever** and an **Adaptive Retrieval Mixer (ARM)** augmentation module. These components work alongside the TSFM backbone, enabling the model to adaptively integrate retrieved information and improve forecasting accuracy. More specifically, the encoder from the pretrained retrieval encoder (*e.g.*, Chronos encoder) is used as the **Retriever Encoder**, which generates embeddings for both the query time series and the time series contexts stored in the retrieval knowledge database. The **Retriever** calculates the Euclidean distance between the query embedding and each stored context embedding in the knowledge base, and then selects the top-$k$ similar candidates based on the smallest distance.

**Retriever**. Formally, given a query context $\mathbf{x}_q$, we first obtain its embedding using the retrieval encoder $f_{\text{enc}}$:

$$\mathbf{e}_q = f_{\text{enc}}(\mathbf{x}_q). \tag{2}$$

Next, the Euclidean distance between the query embedding and each stored embedding in the retrieval knowledge base is calculated:

$$d(\mathbf{e}_q, \mathbf{e}_i) = \|\mathbf{e}_q - \mathbf{e}_i\|_2, \quad \forall i \in \{1, 2, \ldots, n\}. \tag{3}$$

To identify the most relevant time series patterns, the retrieval mechanism selects the top-$k$ candidates with the smallest distance:

$$\mathcal{C} = \text{TopK}_{\text{min}} \left( \{(\mathbf{x}_i, \mathbf{y}_i, d(\mathbf{e}_q, \mathbf{e}_i)) \mid i = 1, 2, \ldots, n\}, k \right), \tag{4}$$

$\text{TopK}_{\text{min}}(\cdot)$ returns the top-$k$ entries ranked by the smallest distance values $d(\mathbf{e}_q, \mathbf{e}_i)$. The retrieved set $\mathcal{C}$ contains the most relevant context-forecast pairs, which are subsequently used to augment the forecasting process.

**Adaptive Retrieval Mixer (ARM)**. To perform forecasting, we develop a novel ARM augmentation module to integrate the projections of the top-$k$ retrieved forecasting horizons with the query time series embedding from the TSFM backbone to enhance prediction accuracy. Each embedding is dynamically weighted by the ARM module and contributes accordingly to the final forecast. Initially, each retrieved forecasting horizon $\mathbf{y}_i$ is encoded independently using a learnable projector:

$$\hat{\mathbf{e}}_i = f_{\text{MLP}}(\mathbf{y}_i), \quad i = 1, 2, \ldots, k \tag{5}$$

where $f_{\text{MLP}}$ is a feedforward network that maps each retrieved sequence into a dense representation of a $d$-dimensional vector. The resulting embeddings are stacked along a new dimension, forming:

$$E_{\text{ret}} = [\hat{\mathbf{e}}_1, \hat{\mathbf{e}}_2, \ldots, \hat{\mathbf{e}}_k] \in \mathbb{R}^{k \times d}, \tag{6}$$

where $d$ is the embedding dimension. To fuse the retrieved information with the query time series representation $\hat{\mathbf{e}}_q \in \mathbb{R}^{1 \times d}$ generated by the TSFM backbone, the two are concatenated into a single representation:

$$E_{\text{concat}} = [\hat{\mathbf{e}}_q; E_{\text{ret}}] \in \mathbb{R}^{(k+1) \times d}. \tag{7}$$

This combined representation is then passed through a Multi-Head Attention (MHA) layer with a residual connection to learn interactions between all the embeddings:

$$E_{\text{att}} = \text{MHA}(E_{\text{concat}}) + E_{\text{concat}}, \tag{8}$$

where $E_{\text{att}} \in \mathbb{R}^{(k+1) \times d}$ represents the contextualized features.

Next, we apply a feed-forward network (FFN) with dropout to further transform these contextualized features, followed by a residual connection to preserve the original information:

$$E_{\text{ffn}} = \text{Dropout}(\text{FFN}(E_{\text{att}})) + E_{\text{att}}. \tag{9}$$

A mixing mechanism is then applied to adaptively balance the contributions of the retrieved sequence representations and the model's original representation. Specifically, a scoring network computes an importance weight for each representation:

$$\alpha = \text{Softmax}(W_g E_{\text{ffn}} + b_g). \tag{10}$$

where $W_g$ and $b_g$ are learnable parameters, and $\alpha \in \mathbb{R}^{(k+1) \times 1}$ denotes the normalized attention weights. The mixed representation is computed as a weighted sum, while a skip connection is applied to preserve the information from the TSFM's pretrained modeling capability:

$$\mathbf{e}_{\text{final}} = \hat{\mathbf{e}}_q + \sum_{i=1}^{k+1} \alpha_i E_{\text{ffn},i}. \tag{11}$$

Finally, the enriched sequence output $\mathbf{e}_{\text{final}}$ is passed through the output projection layer of TSFM to generate the final forecast:

$$\hat{\mathbf{y}}_q = f_{\text{proj}}(\mathbf{e}_{\text{final}}), \tag{12}$$

and we follow the same training objective as the TSFM backbone (see Setup in Section 4.1).

The ARM mechanism enhances forecasting in several key aspects. By leveraging retrieved sequences, the model gains access to additional information, which is particularly valuable when the query context alone is insufficient for accurate predictions. The Multi-Head Attention mechanism enables the model to learn context-aware interactions between the retrieved data and its predictions. Next, the mixing mechanism adaptively determines the importance of each candidate, allowing the model to focus on the most relevant information. Finally, the skip connection ensures that the model's initial predictions are preserved and enriched, maintaining a balance between query knowledge and external augmentation. These designs collectively improve the prediction accuracy and enhance the interpretability of the model, particularly in zero-shot forecasting scenarios.

### 3.3 Pretraining Strategy and Zero-shot Inference

**Pretraining Strategy**. For pretraining, we selectively only train the external parameters of the projector and the ARM augmentation module in TS-RAG based on pre-constructed multi-domain datasets, while keeping all other parameters (the TSFM backbone and the Retrieval Encoder) frozen.

**Zero-shot Inference**. During the zero-shot inference stage, TS-RAG utilizes its pretrained components to generate forecasts without any task-specific fine-tuning. The RAG approach enables TS-RAG to generalize across diverse forecasting tasks by leveraging external knowledge from a broad set of time series domains. Our experiments in Section 4.3 demonstrate the effectiveness of TS-RAG in in-domain, distribution shift, cross-domain, and multi-domain settings.

## 4 Experiments

### 4.1 Experimental Setup

**Datasets and Retrieval Knowledge Base**. For the pretraining dataset, we first uniformly sample 50 million data points from the Chronos pretraining dataset [31] and further uniformly sample a subset

Table 1: Long-term zero-shot forecasting results. Best results are highlighted in **bold**, and second best results are underlined. "—" indicates the datasets were used in pretraining and zero-shot results are not reported. More results are in Appendix B.2 Table 6.

| Methods | TS-RAG$_{\text{Chronos-bolt}}$ | | Chronos-bolt$_B$ | | MOMENT | | TTM$_B$ | | Moirai$_B$ | | TimesFM | | Chronos$_B$ | |
|---|---|---|---|---|---|---|---|---|---|---|---|---|---|---|
| Metric | MSE | MAE | MSE | MAE | MSE | MAE | MSE | MAE | MSE | MAE | MSE | MAE | MSE | MAE |
| ETTh1 | **0.3557** | **0.3624** | 0.3616 | 0.3650 | 0.3920 | 0.4110 | 0.3619 | 0.3710 | 0.3686 | 0.3835 | 0.4254 | 0.3825 | 0.4217 | 0.3806 |
| ETTh2 | **0.2451** | **0.2982** | 0.2517 | 0.2992 | 0.2742 | 0.3327 | 0.2531 | 0.3032 | 0.2547 | 0.3053 | 0.2894 | 0.3233 | 0.2659 | 0.3136 |
| ETTm1 | **0.2906** | **0.3114** | 0.3109 | 0.3185 | 0.3506 | 0.3834 | 0.3152 | 0.3248 | 0.5399 | 0.4322 | 0.3321 | 0.3326 | 0.3935 | 0.3695 |
| ETTm2 | **0.1466** | **0.2231** | 0.1487 | 0.2236 | 0.1703 | 0.2579 | 0.1511 | 0.2405 | 0.1958 | 0.2687 | 0.1703 | 0.2552 | 0.1663 | 0.2522 |
| Weather | **0.1454** | **0.1771** | 0.1525 | 0.1825 | 0.1801 | 0.2384 | 0.1543 | 0.1893 | 0.1711 | 0.1912 | — | — | 0.1897 | 0.2107 |
| Electricity | **0.1120** | **0.2002** | 0.1132 | 0.2004 | 0.1967 | 0.3028 | 0.1715 | 0.2643 | 0.1832 | 0.2814 | — | — | 0.1460 | 0.2237 |
| Exchange rate | **0.0627** | **0.1718** | 0.0673 | 0.1780 | 0.0979 | 0.2059 | 0.0657 | 0.1725 | 0.0663 | 0.1720 | 0.0695 | 0.1802 | 0.0831 | 0.1879 |

of 5 million data points to construct the multi-domain retrieval knowledge base. To facilitate efficient indexing and retrieval, both the pretraining dataset and the retrieval knowledge base are segmented using a predefined context window. This process results in a total of 26 million pretraining data pairs and 2.8 million retrieval knowledge base pairs.

The zero-shot experiments are conducted on widely recognized time series benchmark datasets spanning diverse domains, including ETTh1, ETTh2, ETTm1, ETTm2, Weather, Electricity, and Exchange Rate. Details of these datasets can be found in Appendix A.4. Zero-shot evaluation is performed on the test sets of these datasets, with a data split ratio of 6:2:2 for the ETT datasets and 7:1:2 for Weather, Electricity, and Exchange Rate.

During zero-shot inference, the retrieval knowledge base can be constructed in various ways, including in-domain, distribution shift, cross-domain, and multi-domain settings. We further discuss the setup and impact of different knowledge base choices in Section 4.3.

**Baselines**. In practice, we use Chronos-Bolt, one of the state-of-the-art TSFMs, as the backbone of TS-RAG, as it achieves competitive performance in our evaluations. TS-RAG is designed to be compatible with any general TSFM. While our main experiments primarily use Chronos-Bolt as the backbone due to its strong empirical performance, we also verify TS-RAG's effectiveness with MOMENT [44] in Appendix B.1. For comparison, we also report the zero-shot performance of other TSFMs, including TTM [30], TimesFM [29], Moirai [32], Chronos [31], Chronos-Bolt [31], MOMENT [44], and Time-MoE [48].

**Setup**. Given that TSFMs are typically trained with a fixed forecasting length (*e.g.*, 64 or 96), we maintain this consistency in both pretraining and zero-shot evaluation. The context length is set to 512, and the forecasting length is fixed at 64. We adopt the same forecasting loss as the backbone TSFM; when using Chronos-Bolt as the backbone, we apply the quantile regression loss following its original implementation. Mean Squared Error (MSE) and Mean Absolute Error (MAE) are used as primary evaluation metrics, with detailed definitions provided in Appendix A.5.

## 4.2 Experimental Results for Zero-shot Forecasting

As shown in Table 1, TS-RAG$_{\text{Chronos-Bolt}}$ consistently outperforms other TSFMs, including its backbone Chronos-Bolt, across all datasets, demonstrating its effectiveness in leveraging external patterns to enhance zero-shot forecasting.

Compared to Chronos-Bolt, TS-RAG$_{\text{Chronos-Bolt}}$ achieves an average reduction of 3.54% in MSE and 1.43% in MAE, confirming that the incorporation of retrieved information improves both precision and robustness. Notably, Chronos-Bolt already performs well on the Exchange Rate dataset, achieving an MSE of 0.0673, yet TS-RAG$_{\text{Chronos-Bolt}}$ further reduces the MSE by 6.84%, demonstrating its ability to refine forecasts even when the backbone is highly optimized.

Across each individual dataset, TS-RAG$_{\text{Chronos-Bolt}}$ consistently achieves the lowest MSE and MAE, demonstrating its robustness across diverse time series patterns. Significant performance gains are observed on datasets such as ETTm1 and Weather, where TS-RAG not only outperforms Chronos-Bolt but also surpasses all other TSFMs by a notable margin. This improvement suggests that RAG is particularly effective in datasets with complex temporal dependencies, where incorporating relevant time series patterns from an existing database significantly enhances forecasting accuracy.

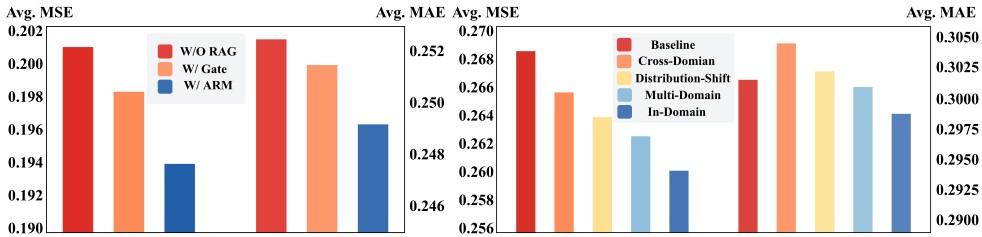

Figure 3: Comparison of average MSE and MAE across augmentation methods (left) and knowledge bases (right). w/o RAG indicates Chronos-Bolt baseline; w/ ARM and w/ Gate refer to TS-RAG with ARM and Gate augmentation modules, respectively. More detailed results are in Appendix B.2

### 4.3 Ablation Studies

**Effectiveness of ARM Module**     The augmentation module plays a critical role in TS-RAG, as it determines how the retrieved forecasting horizons are integrated with the query representation. An ineffective fusion design could limit the benefits of retrieval, leading to sub-optimal forecasting performance. To assess the contribution of the ARM module, we conduct ablation experiments where we replace the ARM with a simpler alternative: a gated fusion module that directly linearly combines the TSFM's forecast with the retrieved forecasting horizon.

We compare the two augmentation modules under the same pretrain-zeroshot setting, keeping all other components fixed. Full results are in Appendix B.2 Table 8. As shown in Figure 3 (left), although the gated fusion variant ("W/Gate") also brings performance improvements over the TSFM baseline, the gains are consistently lower than those achieved by the ARM-based TS-RAG ("W/ARM"). This result highlights the effectiveness of the ARM module in dynamically mixing the contributions of different retrieved patterns as well as the original output representation. The attention-based fusion with residual connection in ARM enables richer interactions and adaptive integration, which proves to be crucial for zero-shot forecasting.

**Impact of Retrieval Knowledge Base**     We evaluate the impact of four retrieval knowledge base configurations on zero-shot forecasting performance, as shown in Figure 3 (right). The configurations include: (1) in-domain, the retrieval knowledge base is built using the training set of the same dataset; (2) distribution shift, the retrieval knowledge base is constructed using the training set from a closely related dataset with a different distribution (*e.g.*, using the ETTh2 training set when evaluating on ETTh1); (3) cross-domain, using different domain data; and (4) multi-domain, using data from multiple domains (as mentioned in the experimental setup, we construct the multi-domain knowledge base using subsets of the pretraining dataset).

The detailed results are in Appendix B.2 Table 7. Across all configurations, TS-RAG improves forecasting performance over the baseline, particularly in MSE, where in-domain retrieval consistently achieves the lowest error. These results suggest that while retrieval generally enhances forecasting, the composition of the retrieval knowledge base plays a crucial role.

**Effectiveness of Longer Forecasting Horizons**     Time Series Foundation Models (TSFMs) typically employ a rolling strategy when forecasting a horizon longer than their pretraining length. In this approach, the model iteratively generates predictions for shorter segments and then rolls forward to forecast the next segment until the full horizon is covered. TS-RAG follows a similar strategy but enhances it with retrieval augmentation. Specifically, for each forecasting step, TS-RAG retrieves the next 64-step forecasting horizon from the retrieval knowledge base, incorporating relevant historical patterns at each iteration until the specified forecasting length is reached.

Table 2 presents the zero-shot forecasting results across multiple datasets, including ETTh, ETTm, Weather, Electricity, and Exchange Rate. The results show that TS-RAG consistently outperforms its backbone model, demonstrating the effectiveness of RAG in extending prediction horizons while maintaining accuracy. The performance gain suggests that leveraging retrieved sequences mitigates error accumulation, a common issue in rolling-based forecasting.

**Effect of Retrieval Lookback Length**     Table 3 presents the effect of different retrieval lookback lengths on zero-shot forecasting performance. Given an input sequence of length 512, we explore

Table 2: Zero-shot forecasting results for extended forecasting horizons across multiple datasets. We report MSE. More results are in Appendix B.2 Table 9.

| Forecasting Length | 96 | | 192 | | 336 | | 720 | |
|---|---|---|---|---|---|---|---|---|
| Methods | w/o RAG | TS-RAG | w/o RAG | TS-RAG | w/o RAG | TS-RAG | w/o RAG | TS-RAG |
| ETTh1 | 0.3859 | **0.3772** | 0.4446 | **0.4306** | 0.4850 | **0.4650** | 0.4841 | **0.4703** |
| ETTh2 | 0.2899 | **0.2812** | 0.3603 | **0.3474** | 0.4045 | **0.3839** | 0.4143 | **0.4017** |
| Electricity | 0.1242 | **0.1226** | 0.1428 | **0.1413** | 0.1613 | **0.1593** | 0.2069 | **0.2050** |
| Exchange rate | 0.0993 | **0.0927** | 0.1926 | **0.1831** | 0.3437 | **0.3157** | 0.8100 | **0.6968** |

Table 3: Long-term zero-shot forecasting results with different retrieval lookback lengths. The best results are highlighted in **bold**, and the second-best results are underlined. MSE is reported here.

| Lookback Length | Metric | ETTh1 | ETTh2 | ETTm1 | ETTm2 | Weather | Electricity | Exchange | Average |
|---|---|---|---|---|---|---|---|---|---|
| 64 | MSE | 0.3540 | 0.2432 | 0.3114 | 0.1502 | 0.1491 | 0.1132 | 0.0678 | 0.1984 |
| 128 | MSE | 0.3572 | 0.2415 | 0.2935 | 0.1494 | 0.1526 | 0.1125 | 0.0674 | 0.1963 |
| 256 | MSE | **0.3539** | **0.2409** | 0.3195 | 0.1518 | 0.1518 | 0.1130 | 0.0662 | 0.1996 |
| 512 | MSE | 0.3557 | 0.2451 | **0.2906** | **0.1466** | **0.1454** | **0.1120** | **0.0627** | **0.1940** |

different retrieval configurations by using only the last 64, 128 or 256 time steps, or the full 512 time steps for retrieval. Across all datasets, longer retrieval lookback windows (256 or 512) yield relatively better performance, suggesting that incorporating a more extended historical context helps retrieve more relevant sequences. This finding demonstrates that retrieving from longer historical sequences generally improves the quality of retrieved sequences, leading to greater forecasting accuracy. However, in some cases, longer is not always better, indicating that excessive retrieval windows may introduce noise or irrelevant information. This suggests the potential for adaptive retrieval mechanisms that allow the retriever to dynamically determine the most suitable retrieval lookback length for each instance.

**Additional Ablation Study** We further conduct additional ablation studies to investigate the impact of retriever configurations, including the number of retrieved sequences, retriever encoder choice, and retrieval distance metrics. The results confirm that while TS-RAG benefits from a moderate number of retrieved sequences, it is robust to the choice of retriever encoder and retrieval distance metric. Detailed results and analysis are provided in Appendices B.3, B.4, and B.5.

## 4.4 Comparison with RAF

To further evaluate the performance of TS-RAG, we compare it with RAF (Retrieval Augmented Forecasting) [53], a recent retrieval-augmented method for zero-shot time series forecasting. As shown in Table 4, we conduct the comparison from two aspects: *effectiveness* and *efficiency*. More implementation details can be found in Appendix A.3.

**Effectiveness** We evaluate the zero-shot forecasting performance of TS-RAG and RAF on seven benchmark datasets. TS-RAG achieves an average MSE of 0.1940, outperforming RAF, which records an average MSE of 0.2320. Moreover, TS-RAG consistently delivers lower errors across all datasets, highlighting its superior generalization capability and robustness.

**Efficiency** We also compare the inference speed of TS-RAG and RAF on the ETTh dataset. TS-RAG significantly outperforms RAF in terms of inference efficiency. Specifically, for the retrieval stage, TS-RAG completes each iteration in just 9.2 ms, whereas RAF requires 3290 ms per iteration. This substantial speed improvement mainly comes from two factors: (1) the use of FAISS for fast nearest-neighbor search, and (2) TS-RAG conducts retrieval over compact embeddings.

## 4.5 Interpretable Forecasting with Case Studies

TSFMs are often used as black boxes, making it difficult to understand how predictions are made. TS-RAG addresses this by providing two key interpretability features: (1) retrieval-as-evidence, which surfaces top-$k$ analogue sequences for each query window, and (2) transparent weighting,

Table 4: Zero-shot forecasting comparison between TS-RAG and RAF. (**Left**) Average MSE and MAE for 512-64 forecasting across seven benchmarks. (**Right**) Average Inference Speed on ETTh datasets. More results are in Appendix B.6.

| Method | Average MSE | Average MAE |
|---|---|---|
| RAF | 0.2318 | 0.2738 |
| TS-RAG (Ours) | **0.1940** | **0.2492** |

| Method | Retrieval | Forward Pass | Total |
|---|---|---|---|
| RAF | 3290 ms/iter | 184 ms/iter | 3474 ms/iter |
| TS-RAG (Ours) | **9.2 ms/iter** | **0.44 ms/iter** | **9.62 ms/iter** |

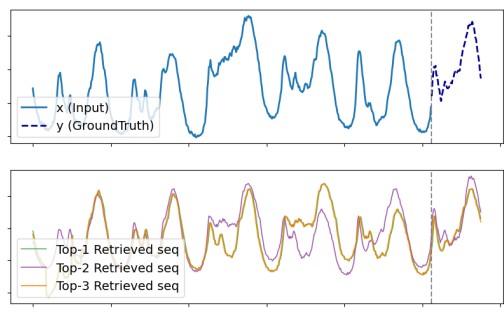

Figure 4: Case study on TS-RAG retrieval (Weather): Given the query time series, the retriever selects relevant historical sequences based on the embedding of the query. The retrieved sequences exhibit strong similarity to the input query in terms of both trend and periodicity.

Figure 5: Case study on TS-RAG retrieval and forecasting (ETTm1): Given the retrieved sequence, the forecasting result with RAG better aligns with the sharp downward trend.

which highlights the most relevant subset of retrieved sequences that have the greatest influence through similarity scores.

We illustrate these capabilities with case studies on retrieval quality and its impact on forecasting performance. Figure 4 shows the retrieval process on the Weather dataset, where the retriever selects highly relevant sequences aligned with both trend and periodicity of the query, serving as explicit evidence for the forecast. Figure 5 presents a case from the ETTm1 dataset, where TS-RAG improves forecast accuracy by leveraging retrieved sequences that exhibit similar sharp downward trends, which the backbone TSFM fails to capture. The retrieved forecasting horizon shown corresponds to the retrieved sequence with the highest weighting, highlighting the model's ability to transparently surface and utilize the most influential retrieved patterns.

These examples demonstrate that TS-RAG not only improves accuracy but also makes its forecasts more interpretable by exposing the key retrieved patterns and their contributions. More case studies are provided in Appendix C.

## 5  Conclusion

In this paper, we introduced TS-RAG, a novel retrieval-augmented forecasting framework designed to enhance the generalization and interpretability of Time Series Foundation Models (TSFMs) in zero-shot forecasting. By integrating retrieval-augmented generation (RAG) with a pretrained retrieval encoder and an Adaptive Retrieval Mixer (ARM) augmentation module, TS-RAG effectively incorporates retrieved relevant patterns to improve forecasting accuracy in previously unseen domains. Extensive empirical evaluations on multiple benchmark datasets demonstrate that TS-RAG can consistently enhance the zero-shot forecasting performance of various TSFMs across diverse domains. Furthermore, we systematically explore the impact of different retrieval configurations, validating TS-RAG as a general and flexible framework for retrieval-augmented time series forecasting.

In summary, TS-RAG establishes a strong foundation for retrieval-augmented time series forecasting, setting up a new frontier for robust and adaptable time series forecasting in dynamic and open world environments. Looking ahead, we aim to 1) explore multimodal extensions of TS-RAG by integrating heterogeneous time series data, such as text data, to further enhance forecasting capabilities; 2) investigate optimization techniques for retrieval ranking in RAG, assessing whether more effective retrieval mechanisms can further boost zero-shot forecasting performance.

## Acknowledgements

The authors gratefully acknowledge the support from Morgan Stanley. Part of this work was conducted during Kanghui Ning's internship at Ant Group.

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

# A Experimental Details

## A.1 TS-RAG Implementation Details

During pretraining, all parameters of the TSFM backbone are frozen, and only the additional parameters introduced by TS-RAG are fine-tuned. The same forecasting loss as the backbone TSFM is adopted; specifically, when using Chronos-Bolt as the backbone, we apply the quantile regression loss following its original implementation. The number of retrieved sequences (top-$k$) is set to 10 by default; however, due to the flexible design of the ARM augmentation module, different values of $k$ can be explored. The model is trained using the AdamW optimizer with a learning rate of 0.0003 and a weight decay of 0.01. The batch size is set to 256, and training is conducted for 10,000 steps. To improve generalization, dropout is applied to certain layers with a dropout rate of 0.2. The training process was conducted on an NVIDIA A6000-48G GPU using TF32 precision. For efficient retrieval, FAISS is used to quickly identify the most relevant sequences from the retrieval knowledge base.

## A.2 TSFM Baseline Introduction

We introduce the baseline models that we choose to compare in the following section:

- **Chronos-Bolt** [31]: Chronos-Bolt is a subsequent version of Chronos, which can handle patch-based and uses decoder representations to generate quantile forecasts across multiple future steps, improving forecast accuracy over Chronos.

- **MOMENT** [44]: MOMENT uses a masking modeling technique for zero-shot forecasting by appending a lookback series with a mask that matches the length of the forecast. It involves pretraining a Transformer encoder model univariately on the "Time Series Pile" datasets, which include a wide variety of time series data.

- **TTM** [30]: TTM pre-trains a compact model based on the lightweight TSMixer architecture. It incorporates adaptive patching, diverse resolution sampling, and resolution prefix tuning to pretrain successfully on a small dataset.

- **Moirai** [32]: Moirai pretrains the Transformer encoder on the "LOTSA" dataset, which includes 27B time points, by masking the forecast horizon of each target channel and performing mask reconstruction.

- **TimesFM** [29]: TimesFM employs a decoder-style attention model and is pre-trained in a univariate manner on a large group of both real-world and synthetic datasets.

- **Chronos** [31]: Chronos is a probabilistic time series foundation model.Chronos tokenizes the input time series in a quantized manner and processes these tokens using the T5 model [56]. Chronos is trained on an extensive corpus of collected and synthetic time series data and has great generalization ability.

- **Time-MOE** [48]: Time-MoE consists of a family of decoder-only time series foundation models with a mixture-of-experts architecture, designed to operate in an auto-regressive manner, enabling universal forecasting with arbitrary prediction horizons and long context lengths.

## A.3 RAF Implementation

Following the original RAF paper, we implement RAF using the Chronos-Base model as the backbone. To ensure a fair comparison, we align both the test set and the retrieval knowledge base with those used in TS-RAG. While the original RAF paper adopts WQL and MASE as evaluation metrics, we report MAE and MSE to maintain consistency with TS-RAG's evaluation protocol.

For the retrieval process, we retrieve the top-10 most relevant sequences, consistent with the TS-RAG setting. Additionally, we implement an RAF variant using the Chronos-Bolt model as the backbone to fully align model configurations. These adjustments ensure that any observed performance differences come solely from the retrieval and augmentation mechanisms.

## A.4 Details of Inference Datasets

We experiment the zero-shot forecasting on the widely adopted Electricity Transformer Temperature (ETT) datasets [14], Weather, Electricity [57], and Exchange Rate from [58]. ETT datasets are comprised of roughly two years of data from two locations in China. The data are further divided into four distinct datasets, each with different sampling rates: ETTh1 and ETTh2 are sampled hourly, and ETTm1 and ETTm2 are sampled every 15 minutes. Every ETT dataset includes six power load features and a target variable: the oil temperature. The Electricity dataset comprises records of electricity consumption from 321 customers and is measured with a 1-hour sampling rate. The Weather dataset contains one-year records from 21 meteorological stations located in Germany. The sampling rate for the Weather dataset is 10 minutes. The Exchange Rate dataset includes the daily exchange rates of eight foreign countries, including Australia, Britain, Canada, Switzerland, China, Japan, New Zealand, and Singapore, ranging from 1990 to 2016.

## A.5 Evaluation Metrics

For evaluation metrics, we use the mean square error (MSE) and mean absolute error (MAE) for zero-shot forecasting. We present the calculations of these metrics as follows:

$$\text{MSE} = \frac{1}{H} \sum_{h=1}^{T} \left( \mathbf{Y}_h - \hat{\mathbf{Y}}_h \right)^2, \qquad \text{MAE} = \frac{1}{H} \sum_{h=1}^{H} \left| \mathbf{Y}_h - \hat{\mathbf{Y}}_h \right|,$$

where $H$ denotes the prediction intervals. $Y_h$ and $\hat{Y}_h$ are the $h$-th ground truth and prediction respectively with $h \in \{1, ..., H\}$. For the evaluation metrics in long-term forecasting, we clarify that the reported metrics are the normalized versions of MAE/MSE. Although we apply global standardization to the data, the information that the scaler used is from training data solely.

## A.6 Efficiency Analysis

**Training Efficiency**  TS-RAG is designed for efficient adaptation. It freezes the TSFM backbone and trains only a lightweight augmentation module (ARM + projection), reducing trainable parameters and improving stability. Retrieval indices are precomputed to avoid redundancy. Training on 26M pairs with 20,000 steps takes around 1 hour on a single NVIDIA A6000 GPU, using TF32 precision and applying dropout.

**Inference Efficiency**  At inference time, TS-RAG introduces a top-$k$ retrieval step. We use FAISS to perform fast nearest-neighbor search over the retrieval knowledge base. On ETTh, retrieval adds 9.2 ms latency per query; the ARM-augmented forward pass adds 0.44 ms. Total inference time is 9.62 ms/query. This overhead is minor and offset by consistent gains (e.g., $-6.84\%$ MSE on Exchange rate), making TS-RAG suitable for real-time use. Notably, compared to RAF, which requires 3.47 seconds per query, TS-RAG achieves over **360×faster** inference speed, highlighting its superior efficiency and practicality.

# B  Additional Results

## B.1 Generalization across Backbones

To verify the generalization ability of TS-RAG, we further evaluate it on the MOMENT [44] backbone, in addition to the Chronos-Bolt results reported in the main text. The same retrieval-augmented framework is applied without modifying the backbone architecture or training recipes, and all retrieval settings follow those used with Chronos-Bolt for fair comparison. *Note that to enable the MOMENT model to perform zero-shot long-term forecasting, we pretrain a prediction head on the same pretraining data as TS-RAG.* **Table 5** reports the results on both backbones. TS-RAG consistently improves zero-shot forecasting performance regardless of the backbone, demonstrating its plug-and-play nature.

Table 5: Zero-shot forecasting results of TS-RAG across different backbones (Chronos-bolt and Moment). Both MSE and MAE are reported. The best results are highlighted in **bold**.

| Backbone | Metric | ETTh1 | ETTh2 | ETTm1 | ETTm2 | Weather | Electricity | Exchange | Average |
|---|---|---|---|---|---|---|---|---|---|
| Chronos-bolt | MSE | 0.3616 | 0.2517 | 0.3109 | 0.1487 | 0.1525 | 0.1132 | 0.0673 | 0.2008 |
|  | MAE | 0.3650 | 0.2992 | 0.3185 | 0.2236 | 0.1825 | 0.2004 | 0.1780 | 0.2525 |
| TS-RAG$_{Chronos-bolt}$ | MSE | **0.3557** | **0.2451** | **0.2906** | **0.1466** | **0.1454** | **0.1120** | **0.0627** | **0.1940** |
|  | MAE | **0.3624** | **0.2982** | **0.3114** | **0.2231** | **0.1771** | **0.2002** | **0.1718** | **0.2492** |
| Moment | MSE | 0.3920 | 0.2742 | 0.3506 | 0.1703 | 0.1801 | 0.1967 | 0.0979 | 0.2374 |
|  | MAE | 0.4110 | 0.3327 | 0.3824 | 0.2579 | 0.2384 | 0.3028 | 0.2059 | 0.3044 |
| TS-RAG$_{Moment}$ | MSE | **0.3823** | **0.2511** | **0.3325** | **0.1552** | **0.1604** | **0.1920** | **0.0775** | **0.2216** |
|  | MAE | **0.4072** | **0.3220** | **0.3738** | **0.2474** | **0.2212** | **0.2994** | **0.1972** | **0.2955** |

Table 6: Full Long-term zero-shot forecasting results across various TSFMs and TS-RAG. Both MSE and MAE are reported. The best results are highlighted in **bold** and second-best results are underlined.

| Backbone | Methods | ETTh1 | ETTh2 | ETTm1 | ETTm2 | Weather | Electricity | Exchange | Average |
|---|---|---|---|---|---|---|---|---|---|
| TS-RAG$_{Chronos-bolt}$ | MSE | **0.3557** | **0.2451** | **0.2906** | **0.1466** | **0.1454** | **0.1120** | **0.0627** | **0.1940** |
|  | MAE | **0.3624** | **0.2982** | **0.3114** | **0.2231** | **0.1771** | **0.2002** | **0.1718** | **0.2492** |
| Chronos-bolt$_B$ | MSE | 0.3616 | 0.2517 | 0.3109 | 0.1487 | 0.1525 | 0.1132 | 0.0673 | 0.2008 |
|  | MAE | 0.3650 | 0.2992 | 0.3185 | 0.2236 | 0.1825 | 0.2004 | 0.1780 | 0.2525 |
| MOMENT | MSE | 0.3920 | 0.2742 | 0.3506 | 0.1703 | 0.1801 | 0.1967 | 0.0979 | 0.2374 |
|  | MAE | 0.4110 | 0.3327 | 0.3824 | 0.2579 | 0.2384 | 0.3028 | 0.2059 | 0.3044 |
| TTM$_B$ | MSE | 0.3619 | 0.2531 | 0.3152 | 0.1511 | 0.1543 | 0.1715 | 0.0657 | 0.2104 |
|  | MAE | 0.3710 | 0.3032 | 0.3248 | 0.2405 | 0.1893 | 0.2643 | 0.1725 | 0.2665 |
| Moirai$_B$ | MSE | 0.3686 | 0.2547 | 0.5399 | 0.1958 | 0.1711 | 0.1832 | 0.0663 | 0.2542 |
|  | MAE | 0.3835 | 0.3053 | 0.4322 | 0.2687 | 0.1912 | 0.2814 | 0.1720 | 0.2906 |
| TimesFM | MSE | 0.4254 | 0.2894 | 0.3321 | 0.1703 | — | — | 0.0695 | — |
|  | MAE | 0.3825 | 0.3233 | 0.3326 | 0.2552 | — | — | 0.1802 | — |
| Chronos$_B$ | MSE | 0.4217 | 0.2659 | 0.3935 | 0.1663 | 0.1897 | 0.1460 | 0.0831 | 0.2380 |
|  | MAE | 0.3806 | 0.3136 | 0.3695 | 0.2522 | 0.2107 | 0.2237 | 0.1879 | 0.2769 |
| Time-MoE | MSE | 0.3623 | 0.2521 | 0.3213 | 0.1565 | 0.1490 | 0.1137 | 0.0851 | 0.2057 |
|  | MAE | 0.3669 | 0.3224 | 0.3340 | 0.2540 | 0.1844 | 0.2026 | 0.2056 | 0.2671 |

## B.2  Ablation Study (Extended Results)

In this section, we present the extended results of the ablation studies reported in the main text. Due to space constraints, some detailed results were omitted or summarized in the main paper. Here, we provide the complete results to facilitate a more comprehensive understanding and reproducibility.

Table 7: Long-term zero-shot forecasting results with different retrieval knowledge bases. The best results are highlighted in **bold**, and the second-best results are underlined.

| Knowledge Base | ETTh1 | | ETTh2 | | ETTm1 | | ETTm2 | |
|---|---|---|---|---|---|---|---|---|
| | MSE | MAE | MSE | MAE | MSE | MAE | MSE | MAE |
| Cross-Domain | 0.3601 | 0.3667 | 0.2466 | 0.2999 | 0.3046 | 0.3240 | 0.1496 | 0.2279 |
| Distribution-Shift | 0.3586 | 0.3647 | 0.2453 | 0.2996 | 0.2993 | 0.3179 | 0.1502 | 0.2271 |
| Multi-Domain | 0.3564 | 0.3633 | **0.2432** | **0.2973** | 0.2971 | 0.3157 | 0.1513 | 0.2277 |
| In-Domain | **0.3557** | **0.3624** | 0.2451 | 0.2982 | **0.2906** | **0.3114** | **0.1466** | **0.2231** |

Table 8: TS-RAG experiment results using different augmentation techniques.

| Method | Metric | ETTh1 | ETTh2 | ETTm1 | ETTm2 | Weather | Electricity | Exchange | Average |
|---|---|---|---|---|---|---|---|---|---|
| TS-RAG$_{ARM}$ | MSE | **0.3557** | **0.2451** | **0.2906** | **0.1466** | **0.1454** | **0.1120** | **0.0627** | **0.1940** |
| | MAE | **0.3624** | **0.2982** | **0.3114** | **0.2231** | **0.1771** | **0.2002** | **0.1718** | **0.2492** |
| TS-RAG$_{Gate}$ | MSE | 0.3575 | 0.2498 | 0.3041 | 0.1473 | 0.1501 | 0.1126 | 0.0663 | 0.1982 |
| | MAE | 0.3640 | 0.2988 | 0.3154 | 0.2235 | 0.1815 | 0.2005 | 0.1768 | 0.2515 |
| Chronos-bolt | MSE | 0.3616 | 0.2517 | 0.3109 | 0.1487 | 0.1525 | 0.1132 | 0.0673 | 0.2008 |
| | MAE | 0.3650 | 0.2992 | 0.3185 | 0.2236 | 0.1825 | 0.2004 | 0.1780 | 0.2525 |

Table 9: Zero-shot forecasting results for extended forecasting horizons (MSE).

| Horizon | Methods | ETTh1 | ETTh2 | ETTm1 | ETTm2 | Weather | Electricity | Exchange | Average |
|---|---|---|---|---|---|---|---|---|---|
| 96 | w/o RAG | 0.3859 | 0.2899 | 0.3323 | 0.1779 | 0.1777 | 0.1242 | 0.0993 | 0.2267 |
| | TS-RAG | **0.3772** | **0.2812** | **0.3141** | **0.1753** | **0.1697** | **0.1226** | **0.0927** | **0.2189** |
| 192 | w/o RAG | 0.4446 | 0.3603 | 0.3838 | 0.2515 | 0.2244 | 0.1428 | 0.1926 | 0.2857 |
| | TS-RAG | **0.4306** | **0.3474** | **0.3688** | **0.2462** | **0.2172** | **0.1413** | **0.1831** | **0.2764** |
| 336 | w/o RAG | 0.4850 | 0.4045 | 0.4374 | 0.3177 | 0.2838 | 0.1613 | 0.3437 | 0.3476 |
| | TS-RAG | **0.4650** | **0.3839** | **0.4153** | **0.3115** | **0.2819** | **0.1593** | **0.3157** | **0.3347** |
| 720 | w/o RAG | 0.4841 | 0.4143 | 0.5285 | **0.4162** | **0.3673** | 0.2069 | 0.8100 | 0.4610 |
| | TS-RAG | **0.4703** | **0.4017** | **0.4935** | 0.4164 | 0.3703 | **0.2050** | **0.6968** | **0.4363** |

## B.3   Sensitivity to the Number of Retrieved Sequences

The impact of varying the number of retrieved sequences ($k$) on forecasting performance is illustrated in Figure 6. The x-axis represents the number of retrieved sequences, while the y-axis shows the corresponding Mean Squared Error (MSE). Across all datasets, increasing $k$ initially leads to a significant decrease in MSE, demonstrating that incorporating additional retrieved sequences helps refine predictions by leveraging retrieved patterns. However, beyond a certain threshold, the improvement plateaus and even decreases slightly in some datasets, indicating diminishing returns as $k$ increases.

Dataset-specific trends further reveal differences in sensitivity to $k$. For instance, ETTm1 and ETTm2 exhibit the most pronounced improvement as $k$ increases, with MSE rapidly declining before stabilizing. This suggests that these datasets benefit significantly from retrieval-augmented inference, likely due to strong temporal dependencies in their historical patterns. ETTh1 and ETTh2 show a similar trend but with a smaller overall reduction in MSE, indicating that while retrieval is beneficial, these datasets may already contain strong intrinsic signals, making additional augmentation less impactful. The Weather, Electricity, and Exchange Rate datasets display a steady decline in MSE with $k$ increasing, but the improvement becomes marginal as $k$ increases further, suggesting that a moderate number of retrieved sequences is sufficient.

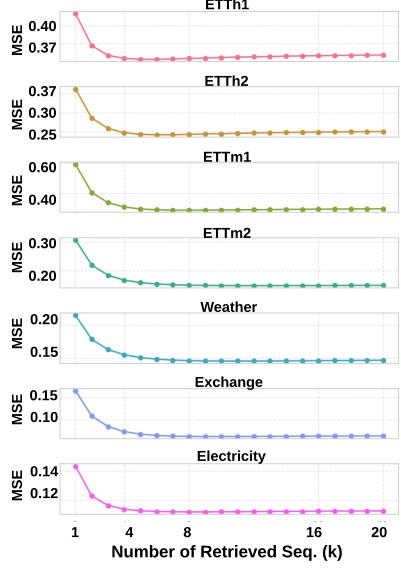

Figure 6: Parameter sensitivity to the number of retrieved sequences (k) on seven zero-shot evaluation datasets.

## B.4   Impact of Retriever Encoder Choice

To further investigate the impact of retriever encoder choice, we conduct an ablation study using pretrained encoders from two additional TSFMs, namely TTM and MOMENT, as alternatives to the

Table 10: Performance comparison of TS-RAG$_{\text{Chronos-bolt}}$ using different **Retriever Encoders** (Chronos, TTM, and MOMENT). Both MSE and MAE are reported.

| Retriever Encoder | ETTh1 | | ETTh2 | | ETTm1 | | ETTm2 | |
|---|---|---|---|---|---|---|---|---|
| | MSE | MAE | MSE | MAE | MSE | MAE | MSE | MAE |
| w/o RAG | 0.3616 | 0.3650 | 0.2517 | 0.2992 | 0.3109 | 0.3185 | 0.1487 | 0.2236 |
| Chronos | 0.3557 | **0.3624** | 0.2451 | 0.2982 | **0.2906** | 0.3114 | 0.1466 | 0.2231 |
| TTM | 0.3556 | 0.3625 | 0.2465 | 0.2993 | **0.2906** | 0.3121 | **0.1452** | **0.2217** |
| MOMENT | **0.3553** | 0.3635 | **0.2437** | **0.2981** | 0.2967 | 0.3164 | 0.1465 | 0.2234 |

Chronos encoder used in our main experiments. TTM employs an MLP-Mixer-like architecture, while MOMENT is based on a Transformer encoder. The Chronos encoder is built on a T5 architecture. All experiments follow the same retrieval-augmented setup for fair comparison.

**Table 10** reports the results. We observe that the performance across the three encoders is largely comparable on all datasets, and no encoder consistently outperforms the others. This suggests that the choice of retriever encoder architecture (among existing TSFMs) has limited impact on the overall performance of TS-RAG.

Table 11: Zero-shot forecasting results under different retrieval distance metrics. Both MSE and MAE are reported.

| Dataset | w/o RAG | | Euclidean | | Cosine | | DTW | |
|---|---|---|---|---|---|---|---|---|
| | MSE | MAE | MSE | MAE | MSE | MAE | MSE | MAE |
| ETTh1 | 0.3616 | 0.3650 | **0.3557** | **0.3624** | 0.3558 | 0.3624 | 0.3606 | 0.3647 |
| ETTh2 | 0.2517 | 0.2992 | **0.2451** | **0.2982** | 0.2465 | 0.2982 | 0.2511 | 0.3001 |
| ETTm1 | 0.3109 | 0.3185 | **0.2906** | 0.3114 | 0.2906 | **0.3111** | 0.3100 | 0.3183 |
| ETTm2 | 0.1487 | 0.2236 | 0.1466 | 0.2231 | **0.1458** | **0.2224** | 0.1489 | 0.2240 |
| Exchange | 0.0673 | 0.1780 | 0.0627 | 0.1718 | **0.0624** | **0.1716** | 0.0665 | 0.1776 |

## B.5 Effect of Retrieval Distance Metrics

To further investigate the effect of retrieval metrics, we conduct additional ablation studies using cosine similarity and DTW distance. For a fair comparison, cosine similarity is calculated over the embeddings generated by the Chronos encoder, while the DTW distance is computed directly in the original time-series space.

From the results in Table 11, we observe that Euclidean and cosine distances achieve very similar performance, both clearly outperforming the baseline setting (i.e., without RAG). In contrast, the DTW distance only achieves modest improvement over the baseline, and in some cases even performs slightly worse. This indicates that embedding-based retrieval methods are more effective and robust for identifying relevant forecasting references than distance measures computed directly in the raw time-series space. Notably, this finding also aligns with prior observations [50] that correlation-based methods are significantly inferior to embedding-based methods for retrieving forecasting references.

## B.6 Comparison between TS-RAG and RAF

As shown in Table 12, the original RAF implementation (RAF$_{\text{Chronos}}$) exhibits significantly inferior performance compared to TS-RAG across all benchmarks. Even when upgrading the backbone of RAF to Chronos-Bolt (RAF$_{\text{Chronos-Bolt}}$), TS-RAG$_{\text{Chronos-Bolt}}$ still consistently outperforms RAF on both MSE and MAE metrics. These results demonstrate the effectiveness of TS-RAG's overall design, which integrates both an optimized retrieval process and an adaptive augmentation module to enhance zero-shot forecasting performance.

Table 12: Full results for zero-shot forecasting comparison between TS-RAG and RAF.

| Method | Metric | ETTh1 | ETTh2 | ETTm1 | ETTm2 | Weather | Electricity | Exchange | Average |
|---|---|---|---|---|---|---|---|---|---|
| RAF$_{Chronos}$ | MSE | 0.4212 | 0.2661 | 0.3927 | 0.1659 | 0.1726 | 0.1377 | 0.0666 | 0.2318 |
| | MAE | 0.3800 | 0.3137 | 0.3695 | 0.2521 | 0.2048 | 0.2189 | 0.1774 | 0.2738 |
| RAF$_{Chronos-bolt}$ | MSE | 0.3660 | 0.2524 | 0.3058 | 0.1477 | 0.1780 | 0.1185 | 0.0632 | 0.2045 |
| | MAE | 0.3710 | 0.3046 | 0.3285 | 0.2281 | 0.2065 | 0.2114 | 0.1725 | 0.2604 |
| TS-RAG$_{Chronos-bolt}$ | MSE | **0.3557** | **0.2451** | **0.2906** | **0.1466** | **0.1454** | **0.1120** | **0.0627** | **0.1940** |
| | MAE | **0.3624** | **0.2982** | **0.3114** | **0.2231** | **0.1771** | **0.2002** | **0.1718** | **0.2492** |

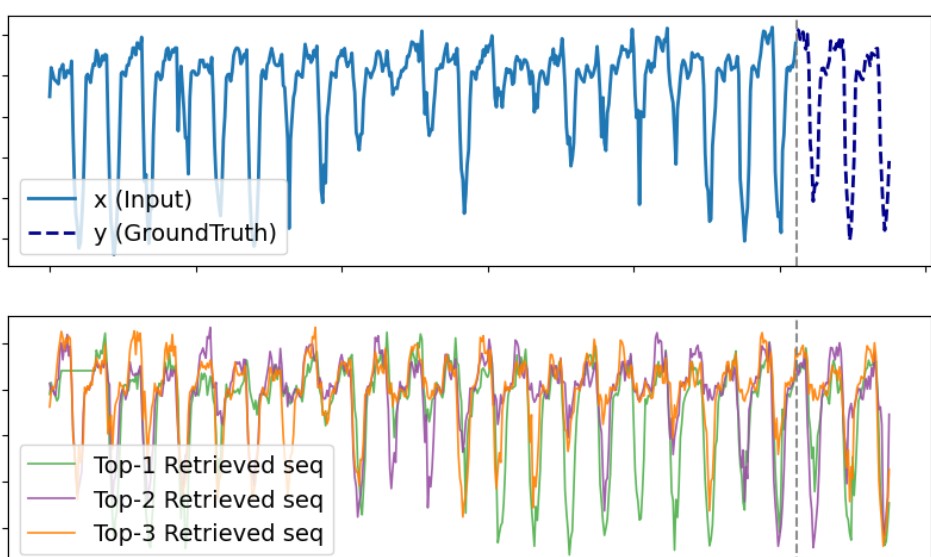

Figure 7: Retrieval results from the ETTh1 dataset.

## C   Showcases

### C.1   Case Studies on Retrieval Effectiveness

Figures 7 and 8 illustrate the retrieval performance of TS-RAG on the ETTh1 and ETTh2 datasets. The retrieval results demonstrate that TS-RAG effectively identifies historical patterns with strong structural similarity to the input, particularly in terms of periodicity and trend dynamics. In ETTh1, the retrieved sequences capture complex fluctuations and local variations, aligning well with the seasonal patterns of the input. Meanwhile, in ETTh2, where the time series exhibits smoother periodicity, the retrieved sequences show almost perfect alignment, indicating the presence of highly consistent cyclic behavior. These results suggest that retrieval augmentation enhances forecasting by leveraging time series patterns that closely match the current context, particularly in datasets with strong seasonal dependencies.

### C.2   Case Studies on Retrieval-Augmented Forecasting

Figures 9 and 10 showcase the impact of retrieval augmentation on forecasting accuracy in the Weather dataset. Figure 9 highlights a situation where the baseline TSFM struggles to capture a sudden trend shift, leading to a significant forecasting error. By incorporating retrieved forecasting horizons, TS-RAG successfully adapts to the trend change. Figure 10 demonstrates how retrieval augmentation enhances peak prediction. The standard TSFM underestimates the upcoming peak, whereas TS-RAG, guided by similar retrieved patterns, generates a more accurate forecast. These case studies illustrate how retrieval-augmented forecasting helps models better adapt to complex temporal patterns, improving robustness in real-world forecasting tasks.

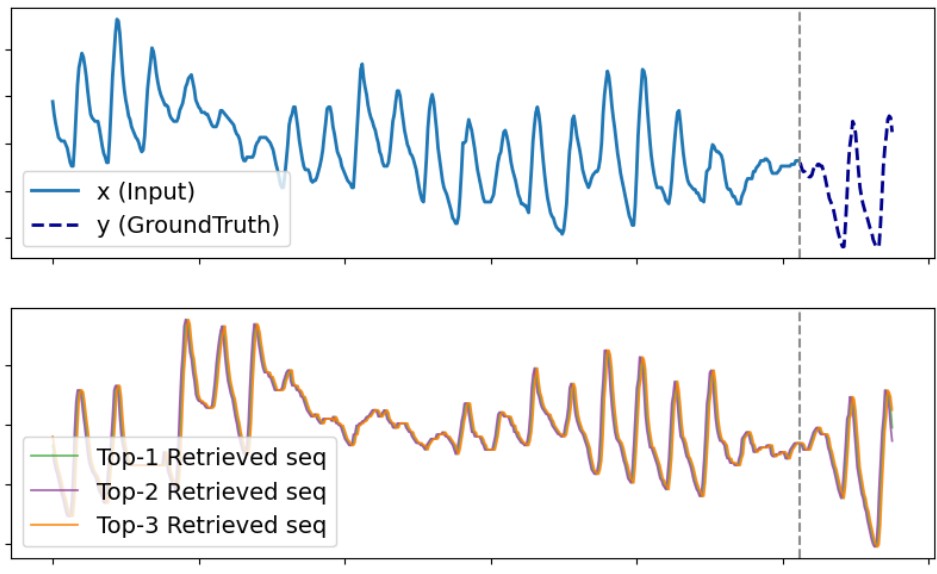

Figure 8: Retrieval results from the ETTh2 dataset.

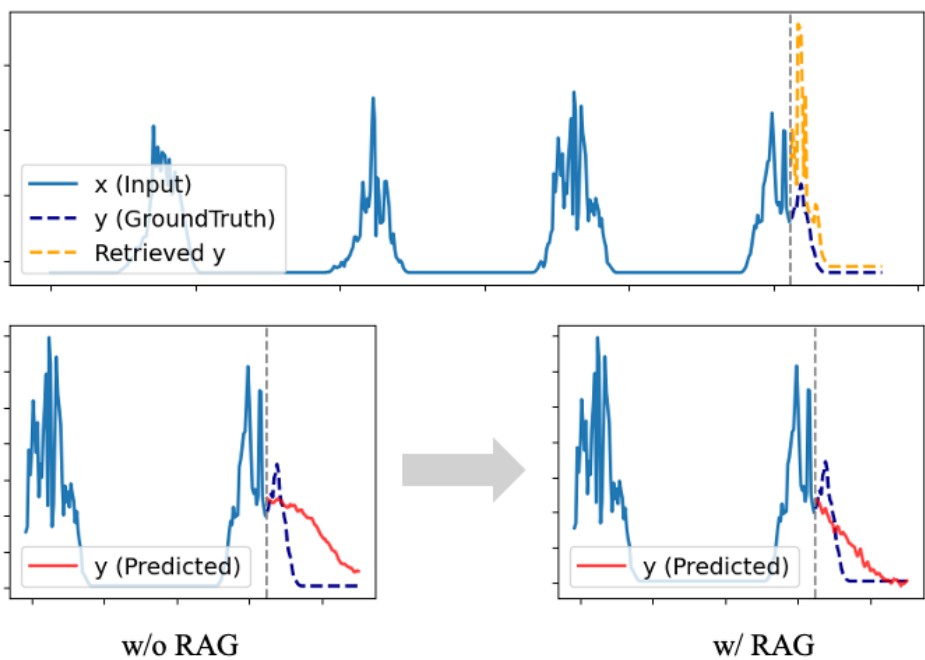

Figure 9: Retrieval-Augment forecasting results from the Weather dataset–example of improving trend adaptation.

## D   Analysis on Time Series Characteristics Influencing TS-RAG Effectiveness

We further investigate which characteristics of time series data make TS-RAG more effective. Specifically, we analyze four statistical properties of the benchmark datasets: **autocorrelation**, **noise ratio**, **volatility**, and **stationarity**, and study how they correlate with TS-RAG's performance improvement over its baseline TSFM.

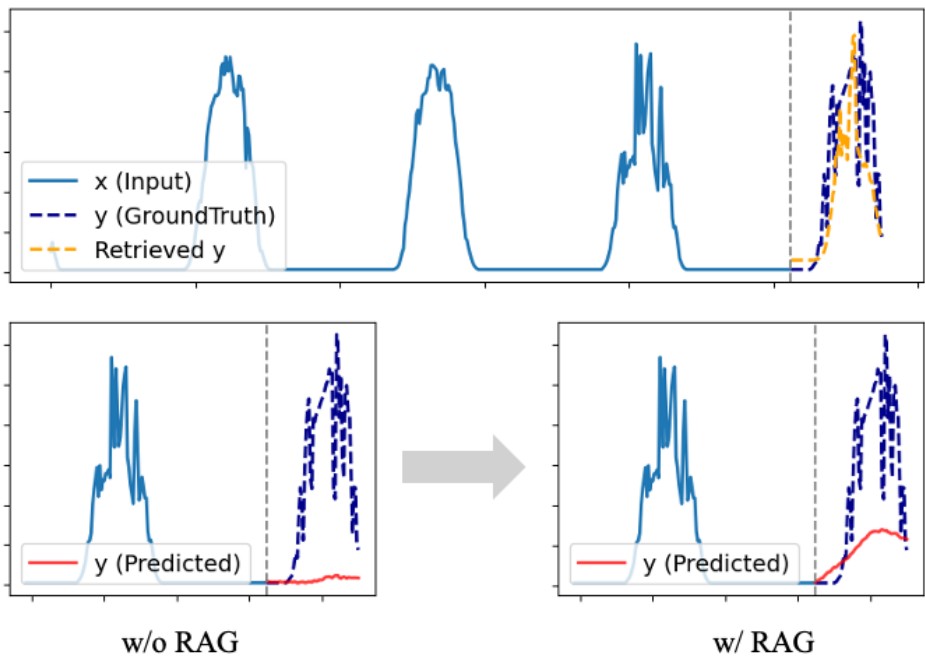

Figure 10: Retrieval-Augment forecasting results from the Weather dataset–example of improving peak prediction.

- **Autocorrelation** measures the strength of temporal dependencies, i.e., how strongly current observations relate to their past values.

- **Noise ratio** reflects the level of irregular fluctuations in the series, computed as the variance ratio of first differences.

- **Volatility** quantifies the variability of the series relative to its mean, computed as the standard deviation divided by the mean value of each sequence. This ratio reflects local fluctuation intensity.

- **Stationarity** estimates how stable the distribution of a time series is over time, approximated by the variance of first differences.

We compute these characteristics for all samples in each dataset and report their average values to represent dataset-level properties. We then calculate the Pearson correlation between each characteristic and the MSE improvement of TS-RAG over the baseline model. Results are summarized in Table 13.

Table 13: Quantitative analysis of dataset characteristics and their correlation with TS-RAG performance gains.

| Dataset | Autocorr. | Noise Ratio | Volatility | Stationarity | MSE Diff |
|---|---|---|---|---|---|
| ETTh1 | 0.7799 | 0.4391 | 3.1655 | 0.1752 | 0.0059 |
| ETTh2 | 0.6070 | 0.7217 | -0.5386 | 0.0843 | 0.0066 |
| ETTm1 | 0.8437 | 0.2915 | -0.9014 | 0.0536 | 0.0203 |
| ETTm2 | 0.6848 | 0.4480 | 17.9827 | 0.0348 | 0.0021 |
| **Correlation** | **0.70** | **-0.55** | **-0.65** | **-0.19** | – |

Our analysis shows that datasets with stronger autocorrelation tend to benefit more from TS-RAG, whereas higher noise levels and greater volatility correspond to smaller improvements. This suggests that the retrieval-augmentation mechanism is particularly effective when temporal dependencies are strong and the underlying patterns are relatively stable.

# E   Limitations

**Limited Modalities.**   While Retrieval-Augmented Generation (RAG) techniques originally stem from the NLP domain, where retrieval often involves textual knowledge, our current implementation focuses solely on time series data due to the lack of available multi-modal datasets. Incorporating rich external information sources such as text or structured metadata could further enhance forecasting performance, particularly in scenarios requiring complex contextual understanding. We leave this as an interesting direction for future work.

**Limited Application Scenarios.**   Our current evaluation focuses on standard public benchmark datasets, which, while diverse, may not fully capture the complexity of real-world forecasting scenarios. Applying TS-RAG to broader application domains, such as finance, healthcare, would further validate its generality and practical value. Exploring these real-world settings remains an important direction for future research.

# F   Broader Impact

This work enhances time series forecasting by leveraging RAG to improve time series foundation model performance. The broader impact of this work can be multifaceted. It may enhance decision-making in critical domains such as finance, healthcare, and environmental monitoring by providing more accurate and reliable forecasts and could lead to better resource allocation, improved patient care, and more effective responses to climate change. No ethical concerns must be considered. The social impacts are significant, as it has the potential to revolutionize our approach to complex time series data and the integration of emerging AI tools, including foundational models. It could change how we analyze and leverage time series data in various fields.

