# OpenReview forum: "TS-RAG: Retrieval-Augmented Generation based Time Series Foundation Models are Stronger Zero-Shot Forecaster"
_NeurIPS.cc/2025/Conference — NeurIPS 2025 poster_

### Official Review · Reviewer_TKBL · 2025-07-02

**Clarity:** 2
**Significance:** 3
**Originality:** 2
**Rating:** 4
**Confidence:** 3

**Summary:**

This work introduces TS-RAG, a retrieval-augmented generation framework for time series forecasting, designed to improve the zero-shot performance and interpretability of Time Series Foundation Models (TSFMs).
TSFMs and LLMs have shown promise in forecasting tasks. However, they often struggle with generalization across diverse domains and suffer from limited adaptability to non-stationary or distribution-shifting time series, especially in zero-shot scenarios. Moreover, interpretability remains limited.
TS-RAG enhances TSFMs by introducing a RAG mechanism tailored for time series forecasting. It retrieves semantically similar time series segments from a pre-built knowledge base using a pretrained retrieval encoder. These retrieved segments are then fused with the current input via an Adaptive Retrieval Mixer (ARM), which dynamically weighs and integrates retrieved knowledge to improve prediction.

**Questions:**

1. While TS-RAG is described as a plug-and-play module, only two backbones (Chronos-Bolt and MOMENT) are evaluated.

2. The ARM module introduces a moderately complex multi-stage mixing mechanism. Gains over simpler fusion methods (e.g., gating) in Table 7 are present but not substantial across all datasets.

**Ethical Concerns:**

["NO or VERY MINOR ethics concerns only"]

**Final Justification:**

The authors have addressed the concerns about diversity, robustness, and generalizability.
I am pleased to see that these discussions will be integrated into the revised version of this submission, which has raised my overall ratings.

**Limitations:**

1. The retrieval knowledge base is constructed from pretraining datasets (e.g., Chronos) without clear discussion on data representativeness, domain coverage, or temporal bias.

2. The method assumes the availability of large-scale time series data for retrieval, which may not be feasible in privacy-sensitive applications (e.g., patient monitoring).

3. There is no mechanism to verify the validity of retrieved sequences or assess if retrievals come from trustworthy sources.

**Paper Formatting Concerns:**

No.

**Quality:**

3

**Strengths And Weaknesses:**

Strengths:

1. The design of TS-RAG is well-motivated and implemented. The combination of a retrieval encoder, a frozen TSFM backbone, and a lightweight ARM (Adaptive Retrieval Mixer) module is effective and modular.
2. The authors evaluate TS-RAG on seven diverse benchmark datasets and conduct multiple ablation studies, cross-domain evaluations, and comparisons with strong baselines, including Chronos-Bolt, MOMENT, and RAF. Besides, TS-RAG shows consistent performance improvements, with up to 6.84% MSE reduction over TSFMs in zero-shot settings.


Weaknesses:

1. While results on Chronos-Bolt and MOMENT are included, more diversity in TSFM architectures (e.g., TimeGPT, Time-MoE) would strengthen generalizability claims.
2. The assumption that 5M–26M samples sampled from Chronos’ dataset are representative may not hold in all domains (e.g., healthcare, rare events).
3. While the ARM is technically promising, the ablation gains over simpler fusion strategies (e.g., gating) are moderate. The overhead of ARM design might not be justified in all settings.

---

> ### Author Rebuttal · Authors · 2025-07-31
>
> ## **W1&Q1: More diversity in TSFM architectures (e.g., TimeGPT, Time-MoE) would strengthen generalizability claims.**
>
> Thank you for your suggestion. We have evaluated the TS-RAG framework on two TSFMs, Chronos-bolt and MOMENT (see page 14, lines 584–591, and Table 4), and observed consistent improvements over their respective baselines. While these results demonstrate the effectiveness of our approach, we agree that testing on more TSFM architectures would strengthen the generalizability claim. To this end, **we additionally evaluated TS-RAG on two strong TSFM backbones: TTM and TimeMoE**. We observed consistent performance improvements on both models, indicating that TS-RAG effectively enhances diverse TSFMs.
>
> Note that we select these two additional backbones because of their superior performance in our experiments. Architectures like Lag-Llama and TimeGPT represent earlier exploration of TSFMs, and TimeGPT is not publicly available due to its commercial use. Thus, we believe the evaluation on these newer and stronger backbones provides a more convincing demonstration of TS-RAG's effectiveness.
>
> | Dataset | TTM w/o RAG | TTM w/ RAG | TimeMoE w/o RAG | TimeMoE w/ RAG |
> |---------|-------------|------------|-----------------|----------------|
> | ETTh1   | 0.3619     | **0.3593**    | 0.3623         | **0.3588**        |
> | ETTh2   | 0.2531     | **0.2442**    | 0.2521          | **0.2437**         |
> | ETTm1   | 0.3152    | **0.3077**    | 0.3213          | **0.3045**         |
> | ETTm2   | 0.1511    | **0.1492**    | 0.1565          | **0.1533**         |
>
> Our experiments cover a diverse set of TSFM architectures. Specifically, Chronos-bolt is a T5 encoder–decoder, MOMENT is a transformer encoder, TTM is an MLP-based architecture, and TimeMoE is a sparse mixture-of-experts transformer. This diversity demonstrates that TS-RAG is not tied to a specific model type but can consistently enhance a wide range of backbone architectures.
>
> ## **W2&L1: The assumption that sampled data is representative in all domains may not hold.**
>
> Thank you for raising this concern. **We would like to emphasize that the random sampling from the Chronos dataset, after the TSMixup process, strongly ensures that the sampled data covers diverse temporal patterns.** More details are as follows:
>
> 1. The Chronos dataset spans a wide range of application domains, including energy, transport, healthcare, retail, web, weather, and finance, with sampling frequencies ranging from 5 minutes to yearly. It also includes synthetic data, further enriching the diversity of temporal patterns.
>
> 2. In addition, a TSMixup process is applied before the final sampling. Base time series from different training datasets are randomly selected and combined through convex interpolation, which further increases diversity. In our work, we then randomly sampled from the Chronos dataset to construct the training and retrieval knowledge bases, ensuring broad coverage and generality.
>
> The strong and consistent performance of the multi-domain retrieval setting (Figure 2, page 8; Table 6, page 15) provides empirical evidence supporting the representativeness and robustness of our chosen knowledge base.
>
> Furthermore, our work provides a system-level evaluation of the retrieval-augmentation mechanism for TSFMs in zero-shot forecasting. We acknowledge that our knowledge base may not fully cover all domains, such as rare events. In practice, however, end users can construct knowledge bases tailored to their specific needs.
>
> ## **W3&Q2: Justification of ARM design.**
>
> Thank you for this comment. We note that the ARM module consistently outperforms the simpler fusion strategy across all evaluated datasets (Table 7 on page 16), which highlights its robustness. In particular, on ETTm1, Weather, and Exchange rate, ARM achieves up to 5.4\% larger gains compared with gating.
>
> While the absolute improvements may appear moderate in some datasets, we believe they are meaningful given the consistent improvement and the challenging zero-shot forecasting setting. Moreover, the consistent gains from a simpler fusion strategy confirm the strength and robustness of our retrieval results, and the additional improvements from ARM further validate the value of its design.
>
> That said, we agree that the augmentation module could be further enhanced in future work, for example by exploring more adaptive fusion strategies or incorporating cross-modal signals to make the retrieval augmentation even more effective.
>
> ## **L2: Large-scale time series retrieval data is unavailable in privacy-sensitive applications.**
>
> Thank you for raising this concern. We agree that privacy-sensitive applications, such as patient monitoring, often pose challenges due to limited access to large-scale in-domain time series data. However, while our experiments (Figure 2 on page 8, Table 6 on page 15) confirm that in-domain retrieval achieves the best performance, **TS-RAG remains robust under other retrieval settings, especially on multi-domain retrieval, the other options for knowledge base are still better than without (w/o) RAG**.
>
> Furthermore, for such privacy-sensitive applications, some strategies can be adapted to enhance the TS-RAG system.
>
> - From the retrieval knowledge base side, synthetic sequences with similar statistical properties (e.g., trend, periodicity) could be generated and indexed into the knowledge base.
>
> - From the retrieval side, although our current retrieval encoder is frozen during pretraining, the deployed versions could allow finetuning over the target private data distribution to better align cross-domain embeddings.
>
> We hope this helps address your concern and will include this discussion in the limitations and future directions.
>
> ## **L3: Validate retrieved sequences come from trustworthy sources.**
>
> Thank you for this point. We agree that TS-RAG currently does not explicitly verify the validity or trustworthiness of the retrieved sequences as this is not the focus of this work. However, this can be addressed with certain modifications. **Since the retrieval database is constructed either by the model creator or by end users, the source, domain and time period of each time series are known.** This information can be presented to the end user as a form of “citation”, providing transparency and a degree of verification.
>
> Furthermore, to enhance the validity of retrieved sequences, possible directions include:
>
> - Applying filtering methods such as anomaly detection or statistical quality checks.
>
> - Introducing confidence scores or trust metrics for retrieved data
>
> We will discuss this as future directions.

---

> > ### Comment · Reviewer_TKBL · 2025-08-08
> >
> > Thanks for the authors' rebuttals.
> > Most of my concerns and questions have been addressed and/or appropriately discussed.
> > I will raise my ratings accordingly.

---

> > > ### Author Response · Authors · 2025-08-08
> > > **Thanks for the reviewer's feedback**
> > >
> > > We sincerely appreciate your thoughtful feedback and are glad that our rebuttal addressed your concerns. Thank you for your willingness to update the rating. Your insights have been valuable in improving the practical utility of the paper. We will incorporate these insights into the next version of the paper.

---

> ### Comment · Area_Chair_ESA5 · 2025-08-05
>
> Dear Reviewer TKBL,
>
> Please help go through the rebuttal and participate in discussions with authors. Thank you!
>
> Best regards,
> AC

---

> ### Author Response · Authors · 2025-08-07
>
> Dear Reviewer TKBL,
>
> We sincerely thank you for acknowledging our contribution and providing constructive feedback. We have made our best effort to address the raised concerns and questions. Considering the limited time for discussion, we would appreciate it if the reviewer could let us know if all the concerns have been satisfactorily resolved. We are also happy to address any remaining or further concerns or questions.

---

### Official Review · Reviewer_gudt · 2025-07-02

**Clarity:** 3
**Significance:** 3
**Originality:** 3
**Rating:** 4
**Confidence:** 3

**Summary:**

This paper presents TS-RAG, a retrieval-augmented generation framework for time series forecasting that enhances the generalization and interpretability of Time Series Foundation Models (TSFMs). It leverages pre-trained time series encoders to retrieve semantically relevant segments from a dedicated knowledge base and employs an Adaptive Retrieval Mixer (ARM) module to dynamically fuse the retrieved patterns with the TSFM’s internal representation.

**Questions:**

1. In the abstract, you claim that TS-RAG "provides desirable interpretability," but the case studies mainly focus on retrieval similarity. Could you elaborate on how the model's internal decision-making process (e.g., attention weights in ARM) enhances interpretability beyond retrieving similar segments?

2. You mention that TS-RAG outperforms existing TSFMs by up to 6.84% across diverse domains. However, the improvement varies across datasets (e.g., smaller gains in ETTh2 vs. larger gains in ETTm1). What specific characteristics of time series data (e.g., stationarity, noise level) make TS-RAG more effective, and could you provide a quantitative analysis of this correlation?

3. The paper states that TS-RAG is compatible with any general TSFM, but experiments primarily use Chronos-bolt and MOMENT. Have you tested it with other architectures like Lag-Llama or TimeGPT-1, and if so, what performance trends were observed? If not, what challenges might arise when adapting TS-RAG to these models?

4. You note that in-domain retrieval achieves the lowest error, but real-world scenarios often lack in-domain data. Could you discuss strategies to optimize TS-RAG's performance when only cross-domain or limited knowledge bases are available?

5. The ablation study on the number of retrieved sequences (k) shows diminishing returns beyond a threshold. However, the optimal k varies by dataset. Is there a dynamic k-selection mechanism that TS-RAG could adopt to adapt to different time series patterns, and what would its design entail?

**Ethical Concerns:**

["NO or VERY MINOR ethics concerns only"]

**Final Justification:**

The authors have thoroughly addressed all reviewer concerns in their rebuttal, providing clear, well-supported responses that directly engage with each point. Their answers are logically structured and backed by empirical evidence (e.g., ablation studies, correlation analyses, and extended experiments with additional architectures), which strengthens confidence in the robustness and generalizability of their work. Notably, they:
- Clarified limitations (e.g., error accumulation in rolling strategies) while proposing actionable future solutions.
- Expanded empirical validation (e.g., testing TS-RAG on TTM and TimeMoE, analyzing data characteristics) to substantiate claims.
- Acknowledged practical challenges (e.g., cross-domain retrieval) and outlined optimization strategies.
- Demonstrated interpretability through attention weights and retrieval dynamics, addressing transparency concerns.

The rebuttal also reflects the authors’ receptiveness to feedback, as they committed to revising the paper to better highlight contributions (e.g., refining the experiment section) and incorporating additional visualizations. While the original submission’s clarity could be improved, the rebuttal effectively bridges this gap with precise explanations and new analyses.

I decide to maintain my initial score. The authors’ comprehensive and constructive responses have resolved all raised issues, and their proposed revisions promise to further enhance the paper’s impact.

**Limitations:**

yes

**Quality:**

3

**Strengths And Weaknesses:**

Strengths:
- Enhances model interpretability through retrieval-as-evidence and transparent weighting of retrieved patterns
- Avoids task-specific fine-tuning, reducing computational overhead while maintaining strong generalization
- Compatible with various TSFMs (e.g., Chronos-bolt, MOMENT), demonstrating flexible plug-and-play design

Weakness:
- Performance depends heavily on the retrieval knowledge base composition
- Long-horizon forecasting relies on rolling strategies, risking error accumulation

---

> ### Author Rebuttal · Authors · 2025-07-31
>
> ## **W1: Performance depends heavily on the retrieval knowledge base composition.**
>
> Thank you for raising this concern. We agree that domain relevance plays an important role. To address this, we conducted comprehensive ablation studies on the choice of knowledge base (Section 4.3, page 7, lines 283–294). The results (Figure 2 in page 8, and Table 6 in page 15) show that **TS-RAG achieves consistently performance improvement across all settings**, including in-domain, multi-domain, distribution-shift, and cross-domain. This demonstrates the robustness of our framework.
>
> Moreover, the ablation study provides practical guidance for real-world applications. It indicates that a domain-relevant knowledge base can yield the ideal results, while multi-domain and cross-domain knowledge bases are still beneficial to improve the forecasting performance when domain-specific data is unavailable.
>
> ## **W2: Long-horizon forecasting relies on rolling strategies, risking error accumulation.**
>
> Thank you for raising this issue. We agree that error accumulation from rolling strategies is a common challenge for current TSFMs and is beyond the main scope of this paper, which focus on improving TSFMs with RAG (with ARM strategies). To address this, we see two possible solutions:
>
> - **Multi-length prediction heads**: Incorporating multi-length prediction heads into the foundation model architecture could help the model better adapt to varying forecast horizons.
>
> - **Longer horizon projection**: Enabling the model to directly project longer future horizons from the hidden embeddings during training and inference could reduce the number of rolling steps required.
>
> We hope this discussion could contribute to the better design and development of TSFMs in the future.
>
> ## **Q1: Discuss the model's internal decision-making process (e.g., attention weights in ARM) enhances interpretability beyond retrieving similar segments.**
>
> Thank you for this insightful question. As noted in Section 4.5 (pages 8–9, lines 327–331), beyond retrieval similarity, **TS-RAG provides interpretability through the transparent attention weights assigned to the retrieved embeddings**. These weights are dynamically generated during training and inference.
>
> For example, given an input query, we can examine the learned weights of the retrieved embeddings. We consistently observe that a few embeddings receive higher weights while others are assigned lower values. This indicates that some retrieved sequences have a stronger influence on the forecasting process than the others, thereby offering interpretability about how the model makes augmentation decisions.
>
> Such transparency can be particularly valuable in real-world applications, where understanding which retrieved patterns affect the forecast helps users build trust in the model’s predictions. While figures are not allowed in the rebuttals, we will add an illustration of attention-based interpretability to the updated manuscript to better explain this.
>
> ## **Q2: What specific characteristics of time series data (e.g., stationarity, noise level) make TS-RAG more effective, and could you provide a quantitative analysis of this correlation?**
>
> Thank you for your thoughtful question. We agree that the characteristics of time series data, such as stationarity, noise level, and the recurrence of patterns, can influence how much TS-RAG improves over baseline TSFMs.
>
> To evaluate this point, we analyzed key characteristics of the seven datasets in the paper. Specially, we measured (1) autocorrelation, which captures the strength of temporal dependencies, (2) noise ratio, reflecting the level of irregular fluctuations, (3) volatility, quantifying the variability relative to the mean, and (4) a stationarity proxy, estimating how stable the series’ distribution is over time. We computed these characteristics using standard statistical measures: lag-1 autocorrelation, variance ratios of first differences (noise ratio), normalized standard deviation (volatility), and variance of first differences as a proxy for stationarity.
>
> | Dataset | Autocorrelation | Noise Ratio | Volatility | Stationarity | MSE Diff |
> |---------|----------------|-------------|------------|--------------|----------|
> | ETTh1   | 0.7799         | 0.4391      | 3.1655     | 0.1752       | 0.0059   |
> | ETTh2   | 0.6070         | 0.7217      | -0.5386    | 0.0843       | 0.0066   |
> | ETTm1   | 0.8437         | 0.2915      | -0.9014    | 0.0536       | 0.0203   |
> | ETTm2   | 0.6848         | 0.4480      | 17.9827    | 0.0348       | 0.0021   |
> | Corr.   | 0.70           | -0.55       | -0.65      | -0.19        | --       |
>
> After we got all the characteristics, we calculate the pearson correlation between the performance improvement and these characteristics. **Our observations suggest that datasets with stronger autocorrelation tend to benefit more from TS-RAG, while higher noise and volatility are associated with smaller improvements**. This supports our hypothesis that retrieval-augmentation mechanism is most effective when temporal dependencies are strong.
>
> ## **Q3: Test TS-RAG with other architectures.**
>
> Thank you for your suggestion. We have evaluated the TS-RAG framework on two TSFMs, Chronos-bolt and MOMENT (see page 14, lines 584–591, and Table 4), and observed consistent improvements over their respective baselines. While these results demonstrate the effectiveness of our approach, we agree that extending the evaluation to more TSFM architectures would further strengthen the generalizability claim. To this end, **we additionally evaluated TS-RAG on two strong TSFM backbones: TTM and TimeMoE**. We observed consistent performance (MSE) improvements on both models, indicating that TS-RAG effectively enhances diverse TSFMs.
>
> Note that we select these two additional backbones because of their superior performance in our experiments. Architectures like Lag-Llama and TimeGPT represent earlier exploration of TSFMs, and TimeGPT is not publicly available due to its commercial use. Thus, we believe the evaluation on these newer and stronger backbones provides a more convincing demonstration of TS-RAG's effectiveness.
>
> | Dataset | TTM w/o RAG | TTM w/ RAG | TimeMoE w/o RAG | TimeMoE w/ RAG |
> |---------|-------------|------------|-----------------|----------------|
> | ETTh1   | 0.3619      | **0.3593**     | 0.3623          | **0.3588**        |
> | ETTh2   | 0.2531      | **0.2442**     | 0.2521          | **0.2437**         |
> | ETTm1   | 0.3152      | **0.3077**     | 0.3213          | **0.3045**         |
> | ETTm2   | 0.1511      | **0.1492**     | 0.1565          | **0.1533**        |
>
> Eventually, our experiments cover a diverse set of TSFM architectures. Specifically, Chronos-bolt is a T5 encoder–decoder, MOMENT is a transformer encoder, TTM is an MLP-based architecture, and TimeMoE is a sparse mixture-of-experts transformer. **This diversity demonstrates that TS-RAG is not tied to a specific model type but can consistently enhance a wide range of backbone architectures.**.
>
> ## **Q4: Discuss strategies to optimize TS-RAG's performance when only cross-domain or limited knowledge bases are available.**
>
> Thank you for raising this point. We agree that the real-world deployment of TS-RAG is truly important.
>
> While our experiments (Figure 2 on page 8, Table 6 on page 15) confirm that in-domain retrieval achieves the best performance, **TS-RAG remains competitive under other retrieval settings (multi-domain, distribution shift and cross-domain)**, especially multi-domain retrieval.
>
> To further optimize TS-RAG in real-world scenarios without in-domain data, several strategies are promising:
>
> - From the knowledge base side, **a prepared multi-domain knowledge which contains diverse temporal patterns** would be helpful in different scenarios. If it’s also not available, **synthetic sequences with similar statistical properties** (e.g., trend,  periodicity) could be generated and indexed into the knowledge base.
>
> - From the retrieval side, although our current retrieval encoder is frozen during pretraining, **the deployed versions could allow finetuning on target data distribution to better align cross-domain embeddings**.
>
> ## **Q5:  Discussion about the potential of Dynamic k-selection mechanism.**
>
> Thank you for this insightful observation. While TS-RAG currently uses a fixed k during inference, we find that the optimal choice is quite stable across diverse datasets. Our analysis shows that selecting k within the range of 8–12 is sufficient to achieve at least 98\% of the optimal performance across all datasets** (see Figure 5, page 16). This indicates that k is a relatively robust hyperparameter for TS-RAG. Nevertheless, we agree that a dynamic selection mechanism is a promising direction.
>
> - One simple approach would be to **set a similarity threshold and retrieve only sequences above that threshold**, thereby avoiding the inclusion of weakly relevant sequences that might dilute the forecast.
>
> - Another possibility is to **use reinforcement learning to learn k-selection policies in an end-to-end manner**. Here, k-selection can be treated as a discrete decision problem: for each input query and its candidate retrieved sequences, a controller predicts the best k from a predefined range, with forecasting accuracy serving as the reward signal.
>
> Such mechanisms are well aligned with the plug-in nature of TS-RAG and may unlock further improvements in generalization. We appreciate the reviewer for raising this valuable point and view it as an exciting direction for future work.

---

> > ### Comment · Reviewer_gudt · 2025-08-05
> > **Thanks for authors' response**
> >
> > Thank you for addressing my concerns. I appreciate the value of your work, and your rebuttal is clear and well-structured. To further strengthen the paper, you might consider revising the experiment section to better emphasize the key contributions.

---

> > > ### Author Response · Authors · 2025-08-05
> > > **Thanks for the reviewer's feedback**
> > >
> > > We sincerely appreciate your thoughtful questions and constructive feedback, as well as your recognition of the value of our work. We are glad that our rebuttal addressed your concerns clearly. In the next version, we will incorporate additional experiments and refine the presentation to further highlight our contributions.

---

### Official Review · Reviewer_6MH9 · 2025-07-03

**Clarity:** 3
**Significance:** 3
**Originality:** 3
**Rating:** 4
**Confidence:** 3

**Summary:**

The paper introduces TS-RAG, a retrieval-augmented framework that plugs time-series foundation models into a fast nearest-neighbour search to improve zero-shot forecasting. It first embeds the query’s recent history with a frozen encoder, retrieves the k most similar past windows, and feeds the corresponding true future segments into an Adaptive Retrieval Mixer (ARM). ARM learns attention weights that blend these historical outcomes with the backbone model’s own prediction, while keeping both the backbone and the retriever fixed. Evaluated on seven standard benchmarks, TS-RAG cuts mean-squared and absolute errors and delivers lower inference latency than a prior retrieval baseline.

**Questions:**

Q1. Nearest neighbours are ranked only by Euclidean distance in embedding space. This may fail under temporal misalignment or regime shifts. Report results for at least one stronger retrieval scheme: cosine similarity, DTW-aware embeddings, or a learned metric layer.

Q2. TS-RAG combines diverse historical futures but only reports MSE/MAE. We do not know whether the method yields well-calibrated predictive distributions. Add at least one proper probabilistic metric comparing TS-RAG to the backbone and RAF.

Q3. The retrieval index is built from the training split of the same benchmark, so windows that are temporally close but still overlap in pattern with the test set might be retrieved. Please report the degree of overlap between the index and each test split, and re-run TS-RAG with an index built strictly from other datasets or from earlier, non-overlapping time blocks if necessary.

**Ethical Concerns:**

["NO or VERY MINOR ethics concerns only"]

**Final Justification:**

My concerns have been thoroughly addressed through the rebuttal, so I raise my score accordingly.

**Limitations:**

1. Explain whether the retrieval component could amplify historical biases (e.g., over-representing dominant markets) and propose mitigation (balanced retrieval sampling, debiasing losses)

**Quality:**

3

**Strengths And Weaknesses:**

**Strengths:**

S1. TS-RAG shows how to graft the RAG idea onto generic TSFMs, letting a forecaster pull in semantically similar historical windows and use them during inference.

S2. The proposed ARM block learns attention weights between the query embedding and each retrieved future, then gates them against the backbone forecast, yielding consistent gains over simpler fusions.

S3. Plug-and-play design can generalise across backbones and retriever encoders. Only the projector and ARM are trained, and both the backbone TSFM and the retriever encoder remain frozen.

**Weaknesses:**

W1. The retriever ranks neighbours only by Euclidean distance, which only focuses on the point-level similarity. No exploration of DTW, cosine, or learned metrics, so robustness to misalignment/regime shift is unknown.

W2. Only deterministic point-forecast errors (MSE/MAE) are reported; probabilistic scores are absent, so uncertainty calibration is untested.

W3. Best results come when the knowledge base is built from the training split of the same dataset. Cross-domain retrieval is weaker, showing the method needs closely matched data at inference time.

---

> ### Author Rebuttal · Authors · 2025-07-31
>
> ## **W1&Q1: Report results for at least one stronger retrieval scheme: cosine similarity, DTW-aware embeddings, or a learned metric layer.**
>
> Thanks for your insightful comment. To further investigate the effect of retrieval methods, we conducted additional ablation studies using **cosine similarity and DTW distance**. For a fair comparison, we calculated cosine similarity over the embeddings generated by Chronos encoder, while the DTW distance was computed in the original time series space.
>
> The results (MSE/MAE) are shown as below,
>
> | Dataset   | w/o RAG        | Euclidean      | Cosine         | DTW            |
> |-----------|----------------|----------------|----------------|----------------|
> | ETTh1     | 0.3616 / 0.3650 | **0.3557** / **0.3624** | 0.3558 / 0.3624 | 0.3606 / 0.3647 |
> | ETTh2     | 0.2517 / 0.2992 | **0.2451** / **0.2982** | 0.2465 / 0.2982 | 0.2511 / 0.3001 |
> | ETTm1     | 0.3109 / 0.3185 | **0.2906** / 0.3114 | 0.2906 / **0.3111** | 0.3100 / 0.3183 |
> | ETTm2     | 0.1487 / 0.2236 | 0.1466 / 0.2231 | **0.1458** / **0.2224** | 0.1489 / 0.2240 |
> | Exchange rate | 0.0673 / 0.1780 | 0.0627 / 0.1718 | **0.0624** / **0.1716** | 0.0665 / 0.1776 |
>
> From these results, we observe that **Euclidean and cosine distances achieve very similar performance**, both clearly outperforming the baseline setting, i.e., without (w/o) RAG. In contrast, **DTW distance only achieves modest improvement** over the baseline setting. In some cases, DTW even perform slightly worse than the baseline setting, which suggests that it does not work well to retrieve the sequences that are beneficial to forecasting. Notably, this finding is consistent with the results in [1], which report that correlation-based methods are significantly inferior to the embedding-based methods for retrieving forecasting references.
>
> We provide two empirical explanations for this finding:
>
> - Retrieval directly in the raw time series space tends to introduce more noise, making it harder to identify sequences that are truly useful for forecasting.
>
> - The retriever encoders we use are pretrained on forecasting tasks, which equips them with the ability to identify sequences with similar future dynamics, since sequences with similar horizons naturally have closer embeddings.
>
> Based on these experiments and the results in Table 10 (in Appendix B.5, page 17) of our paper, we summarize the effect of retrieval methods as follows:
>
> 1. The choice between Euclidean and cosine distance has marginal impact on performance.
>
> 2. Using different pretrained encoders also produces only a relatively small impact.
>
> 3. The embedding-based methods are consistently better than DTW distance in the retrieval-augmentation task.
>
>
> ## **W2&Q2: Add at least one proper probabilistic metric comparing TS-RAG to the backbone and RAF.**
>
> Thanks for the comment, we followed the settings in our experiment and report the probabilistic metric (WQL) for Chronos-bolt, RAF [2], and ours for comparison:
>
> | Dataset | TS-RAG (Chronos-Bolt) | RAF (Chronos-Bolt) | Chronos-Bolt |
> |---------|------------------------|--------------------|--------------|
> | ETTh    | **0.123**                  | 0.131              | 0.134        |
> | ETTm    | **0.105**                  | 0.111              | 0.116        |
> | Exchange rate | **0.015**                  | 0.016              | 0.017        |
>
> TS-RAG consistently achieves the best probabilistic forecasting performance compared to other strong baselines on various datasets.
>
> ## **W3: In-domain knowledge base requirement at inference time.**
>
> Thank you for pointing this out. We agree that domain relevance plays an important role for TS-RAG. To address this, we conducted comprehensive ablation studies on the choice of knowledge base (in section 4.3, page 7, lines 283-294). The results (Figure 2 in page 8, and Table 6 in page 15) show that **TS-RAG achieves consistent MSE improvements across all settings**, including in-domain, multi-domain, distribution-shift, and cross-domain. This demonstrates the robustness of our framework.
>
> In addition, the ablation study provides practical guidance for real applications. It shows that using a domain-relevant knowledge base leads to the best results, while multi-domain or cross-domain knowledge bases can still deliver meaningful gains when domain-specific data is not available.
>
> ## **Q3: Concern about overlap between test split and retrieval index.**
>
> Thank you for the comment. To avoid retrieving sequences that overlap with the test queries, **we build the knowledge base strictly from the training split and exclude the validation and test splits. This design ensures that all retrieved windows come from non-overlapping time blocks.**
>
> For example, in ETTh1 dataset, **the timestamp cutoff for the knowledge base is 2017/6/25 23:00, while the test set starts from 2017/10/2 15:00**. This approach ensures that the retrieved sequences have no overlap with the input queries. We hope this clarification helps address your concern.
>
> ## **L1: Whether the retrieval component could amplify historical biases.**
>
> Thank you for raising this concern. We agree that retrieval could potentially amplify historical biases if the knowledge base over-represents certain markets or patterns. Nevertheless, as noted on page 6, lines 206–214, in our current design **the attention-based ARM module dynamically leverages multiple retrieved sequences to augment forecasting. This mechanism helps the TS-RAG framework remain robust even when the knowledge base may contain some bias.**
>
> That said, we acknowledge this as a potential limitation in broader real-world applications, and we will include the following discussion in the limitations section.
>
> To mitigate such risks, we see two promising directions:
>
> - **Balanced retrieval sampling**: Applying balanced retrieval sampling to improve the diversity of retrieved sequences
>
> - **Noise injection**: Add controlled noise into the retrieval results to reduce over-reliance on dominant patterns. Notably, [3] has observed benefits of noise for RAG systems.
>
> [1] Liu, Jingwei et al. Retrieval-augmented diffusion models for time series forecasting,Advances in Neural Information Processing Systems 37 (2024).
>
> [2] Kutay Tire et al. Retrieval Augmented Time Series Forecasting, Association for the Advancement of Artificial Intelligence (2025).
>
> [3] Florin Cuconasu et al. The Power of Noise: Redefining Retrieval for RAG Systems, Proceedings of the 45th International ACM SIGIR Conference on Research and Development in Information Retrieval (2024).

---

> > ### Comment · Reviewer_6MH9 · 2025-08-05
> >
> > Thank you for your response and the additional experiments. My concerns have been thoroughly addressed.
> >
> > I will update my score accordingly. Please consider incorporating the results and justifications into the paper to strengthen its quality.

---

> > > ### Author Response · Authors · 2025-08-05
> > > **Thanks for the reviewer's feedback**
> > >
> > > We sincerely appreciate your thoughtful comments and are glad that our responses have addressed your concerns. Thank you as well for your willingness to update the score. We will incorporate these results and justifications into the next version of the paper to further strengthen its quality.

---

> ### Comment · Area_Chair_ESA5 · 2025-08-05
>
> Dear Reviewer 6MH9,
>
> Please help go through the rebuttal and participate in discussions with authors. Thank you!
>
> Best regards,
> AC

---

### Official Review · Reviewer_yyAV · 2025-07-04

**Clarity:** 3
**Significance:** 3
**Originality:** 4
**Rating:** 5
**Confidence:** 5

**Summary:**

This paper proposes TS-RAG, a novel retrieval-augmented generation framework for time series forecasting, with a focus on addressing the generalization and interpretability challenges of Time Series Foundation Models. The key innovation lies in combining a pretrained retriever encoder with an Adaptive Retrieval Mixer module to dynamically integrate retrieved relevant time series segments to improve zero-shot forecasting performance without fine-tuning. The empirical study over seven benchmark datasets demonstrates the superiority of TS-RAG over state-of-the-art TSFMs for zero-shot forecasting tasks. Overall, this work establishes a strong foundation for retrieval-augmented time series modeling.

**Questions:**

Q1: Please provide more explanations for the results in Figure 2, knowledge bases (right).
Q2: While TS-RAG effectively retrieves time series segments to enhance zero-shot forecasting performance of TSFMs, I wonder whether incorporating external modalities (e.g., text or structured metadata) could further enhance the forecasting performance. (I understand this may not be the focus of this paper, but providing some insights or discussions would be helpful.)

**Ethical Concerns:**

["NO or VERY MINOR ethics concerns only"]

**Final Justification:**

After reading the authors’ rebuttal and other reviewers’ comments, I insist on my original evaluation towards this paper. The main idea of incorporating RAG into time series is interesting. The overall quality of this paper is good. It is a good paper and I recommend it to be accepted.

**Limitations:**

See weakness.

**Quality:**

3

**Strengths And Weaknesses:**

Strength:
1.	This paper adapts the retrieval-augmented generation paradigm from NLP to time series by incorporating an Adaptive Retrieval Mixer module that can dynamically fuse retrieved historical patterns with the query, and enhance model generalization without task-specific fine-tuning. This design fills a critical gap in current TSFM literature where adaptation and external knowledge incorporation remain limited.
2.	This paper is well-written, and the empirical results are quite solid. Extensive zero-shot time series forecasting experiments have been conducted over 7 diverse public benchmark datasets under the various retrieval knowledge base settings of in-domain, cross-domain, and multi-domain. TS-RAG shows consistent improvements in MSE and MAE across all datasets, with different TSFMs and settings.
3.	Ablation studies on module design choices and retrieval configurations are provided, and the results are convincing to validate the effectiveness and robustness of the TS-RAG model design. Case studies are also provided to illustrate how retrieved sequences align and work with input queries, providing explainability/insight for model reasoning.
4.	Code has been provided and properly documented to improve the reproducibility.

Weaknesses:
1.	It would be better to provide more explanations for the results in Figure 2, knowledge bases (right).
2.	Some implementation details, such as hyperparameter choices for ARM (e.g., MHA dimension) and retrieval database construction strategies, are provided in the appendices. It would be better to provide selected key information in the main text to improve clarity.

---

> ### Author Rebuttal · Authors · 2025-07-31
>
> ## **W1&Q1: Please provide more explanations for the results in Figure 2, knowledge bases (right).**
>
> Thank you for the suggestion. We provide clarification on the settings of in-domain, distribution shift, and cross-domain retrieval in the “Impact of Retrieval Knowledge Base” section (page 7, lines 283–294), along with the corresponding main results. **In the updated version, we will also include a more detailed analysis in the appendix to further support our findings.**
>
> Across different retrieval knowledge base configurations, each choice leads to improvements over the baseline without RAG. Among them, the in-domain setting yields the highest gain, followed by the multi-domain setting, with the distribution-shift and cross-domain settings performing slightly lower. These results suggest that the relevance between candidate sequences in the knowledge base and the input query plays a crucial role in the TS-RAG framework: the higher the relevance, the greater the improvement can be achieved through retrieval augmentation. This finding provides guidance for building the knowledge base in real applications: a domain-relevant knowledge base is the ideal choice. When the domain-relevant knowledge base is not available, a multi-domain knowledge base containing diverse time series patterns can still be highly beneficial.
>
> ## **W2: Some implementation details are provided in the appendices. It would be better to provide selected key information in the main text to improve clarity.**
>
> Thank you for your suggestion, we will include the essential implement details in the experimental setup section in the final version.
>
> ## **Q2: Whether incorporating external modalities (e.g., text or structured metadata) could further enhance the forecasting performance.**
>
> Thank you for raising this insightful point. We agree that incorporating external modalities could further enhance forecasting performance. In this work, we propose TS-RAG, a retrieval-augmented forecasting framework that has been shown to effectively improve the performance of TSFMs, and it also opens the door to integrating additional modalities in the future.
>
> Recent studies have demonstrated that multimodality, particularly the integration of textual information, can significantly improve time series forecasting [1, 2]. Nonetheless, as noted in [3], the effectiveness of such approaches depends on factors such as model capacity, alignment strategies, and the quality of external data. We believe that a strong and comprehensive forecasting system should leverage both retrieval augmentation and external modalities, though further exploration is required. Particularly, we think some questions are worth to explore:
>
> - **When and how to use external modalities**: External information may not always be beneficial. Developing mechanisms to determine when and how to effectively incorporate them is essential.
>
> - **Multimodal RAG**: Extending RAG to handle heterogeneous modalities raises questions about how to model different modalities within a shared embedding space and how to efficiently leverage retrieved multimodal data to augment forecasting.
>
> We appreciate your thoughtful suggestion and plan to leave these directions for future work.
>
> [1] Liu, Haoxin, et al. ”Time-MMD: Multi-Domain Multimodal Dataset for Time Series Analysis.” Advances in Neural Information Processing Systems 37 (2024)
>
> [2] Andrew R. Williams, et al. ”Context is Key: A Benchmark for Forecasting with Essential Textual Information.” Advances in Neural Information Processing Systems 37 (2024)
>
> [3] Zhang, Xiyuan, et al. ” Does Multimodality Lead to Better Time Series Forecasting? ” Arxiv (2025)

---

### Decision · Program_Chairs · 2025-09-17

**Decision:**

Accept (poster)

**Comment:**

This paper introduces TS-RAG, a retrieval-augmented framework for time series forecasting that enhances the generalization and interpretability of Time Series Foundation Models. The key contribution lies in integrating retrieved historical patterns with a pretrained backbone via an Adaptive Retrieval Mixer, which improves zero-shot forecasting without fine-tuning. Before the rebuttal, the reviewers raised some concerns, such as reliance on knowledge base quality, limited diversity in backbone models, and relatively modest gains of ARM over simpler strategies. During the rebuttal, the authors have satisfactorily addressed these concerns and all the reviewers are positive with this paper. Therefore, I would like to recommend accepting this paper.